# Controlling Structured Explanations via Shapley Values

## Abstract

Structured explanations elucidate complex feature interactions of deep networks, promoting interpretability and accountability. However, existing work primarily focuses on post hoc diagnostic analyses and does not address the fidelity of structured explanations during network training. In contrast, we adopt a Shapley value-based framework to analyze and regulate structured explanations during training. Our analysis shows that valid subexplanation counts in structured explanations of Transformers and CNNs strongly correlate with each model's feature interaction strength. We also adopt a Shapley value-based multi-order interaction regularizer and experimentally demonstrate on the large-scale ImageNet and fine-grained CUB-200 datasets that this regularization allows the model to actively control explanation scale and interpretability during training.

## 1 Introduction

The growing demand for explainable vision models necessitates explanations beyond conventional saliency maps, which inherently oversimplify the complex interactions among visual cues that the model actually processes. In contrast, structured explanations—such as Structured Attention Graphs (SAG) (Shitole et al., 2021)—systematically capture both local saliency and global reasoning pathways. By explicitly capturing how visual details interact with broader contextual cues, these methods yield more informative and interpretable explanations, which is critical for high-stakes applications where transparency is essential (e.g., medical imaging and autonomous driving). Beyond improving interpretability, structured explanations can enhance robustness, modularity, and user understanding, enabling better reasoning about model predictions and counterfactuals (Shitole et al., 2021).

While structured explanations offer valuable insights into model reasoning, they remain purely diagnostic and do not support the integration of explanatory guidance into the training process. This limitation reveals a critical gap: without incorporating explanatory feedback during learning, models can achieve high performance while relying on spurious or opaque reasoning. To ensure that predictions are not only accurate but also grounded in meaningful, human-aligned rationale, it is essential to directly integrate the explanation objectives into training—moving beyond post hoc analysis.

In this paper, we characterize structured explanations of various models within a theoretical framework based on Shapley values (Deng et al., 2022), and leverage this foundation to analyze how interaction structure relates to post-training explanations. Originating in game theory, Shapley values provide a principled method to fairly allocate a total payoff among a set of cooperating agents based on their individual contributions. Our first contribution identifies a principled connection between a popular metric for characterizing structured explanations—the count of minimal sufficient subexplanations (Jiang et al., 2024)—and the strength of feature interactions as quantified by Shapley values. Our experiments across a diverse set of models demonstrate a strong correlation between the subexplanation metric and feature-interaction strength: models with stronger feature interactions across mid-range spatial contexts in the input image also tend to yield higher subexplanation counts. By establishing this relationship at different granularity levels (or scales) of explanations—where scale refers to the spatial extent of the input region needed to preserve a valid explanation—we also provide insight into how various models integrate features during inference, suggesting that feature-interaction strength could serve as a lever to influence the decision-making pathways.

Motivated by the observed correlation between interaction structure and explanation patterns, our second contribution enables explicit control over the scale and interpretability of structured expla-

nations—and by extension, aspects of model behavior—during training. To ensure the explanations align with specific XAI objectives, we adopt the differentiable feature-interaction loss from Deng et al. (2022) as a regularization term in end-to-end learning. In this way, Shapley-based interaction strength serves not only as a post-training diagnostic tool but also as an optimization constraint during training. While prior work applied this loss mainly to CNNs and only to analyze the CNN's representational bottleneck, it did not study how modifying interaction strength affects explanation scales or reasoning patterns. Here, we extend prior work to a range of models and evaluate the effects of this regularization using multiple interpretability metrics based on SAGs (Shitole et al., 2021), subexplanation counts (Jiang et al., 2024), saliency distributions, and ProtoTrees (Nauta et al., 2021). Our experiments demonstrate that targeted manipulation of interaction strength effectively modifies structured explanation scales, shifts subexplanation patterns, and influences model reasoning behaviors observed across architectures. Importantly, our goal is not to characterize or optimize the accuracy–explainability tradeoff, but to show how explanation structure can be systematically regulated while maintaining competitive accuracy, leaving a study of such tradeoffs to future work.

The rest of the paper proceeds as follows: Sec. 2 reviews related work. Sec. 3 covers the theory behind Shapley values. Sec. 4 presents our correlation analysis between interaction strengths and structured explanation scales. Sec. 5 details the interaction-based regularization and evaluates its impact on post-training explanations and model behavior.

## 2 RELATED WORK

This section reviews related explainability frameworks and work on Shapley values.

**Attribution maps**, or heatmaps, explain model predictions by assigning importance scores to input features. Gradient-based methods (Bach et al., 2015; Selvaraju et al., 2017; Sundararajan et al., 2017; Zeiler & Fergus, 2014; Adebayo et al., 2018; Nie et al., 2018; Zimmermann et al., 2021) compute the heatmaps by backpropagating gradients to input features, but fail to capture feature interactions. Attention-based methods (Abnar & Zuidema, 2020; Vig et al., 2021; Clark et al., 2019; Vaswani et al., 2017) estimate the importance of different input parts. The resulting heatmap serves as an explanation by visually highlighting input regions that the model's attention was focused on during processing, but does not reveal the model's full reasoning process.

**Structured explanations** have been advocated for more effective model interpretations (Ribeiro et al., 2018; Geiger et al., 2021; Janizek et al., 2021; Schnake et al., 2021; Feng & Steinhardt, 2024). Sufficient input subsets (Carter et al., 2019) and SAGs (Shitole et al., 2021) generate explanations as minimal subsets of image patches sufficient to replicate the model's output on the entire image. SAG scale can be quantified with valid subexplanation counts (Jiang et al., 2024), i.e., by counting feature combinations that preserve model confidence above a threshold. While subexplanation counts highlight behavioral differences between transformers and CNNs, they fail to reveal the underlying principles of these differences and offer no practical guidance for model training. Prior work has used explanation-related regularization to encourage models to follow human-defined or semantic cues (Ross et al., 2017; Ismail et al., 2021; Plumb et al., 2020), but these methods generally focus on isolated attributions and do not reveal how explanation scales relate to model reasoning. Our work addresses both of these limitations by showing structured explanation scales are essentially characterized by multi-order Shapley interaction strengths, enabling our formulation to be seamlessly incorporated as an explanatory regularization loss in end-to-end training.

**Shapley values** (Roth, 1988; Winter, 2002; Shapley, 1953; Grabisch & Roubens, 1999; Sundararajan et al., 2020; Covert et al., 2023; Ancona et al., 2019b; Tsai et al., 2023; Jethani et al., 2022; Chen et al., 2023b;a; Covert & Lee, 2021) quantify individual feature contributions to the model's output by averaging marginal effects over all possible feature subsets. CS-Shapley (Schoch et al., 2022) computes class-specific Shapley values to address class imbalance. Sobol indices (Fel et al., 2021; Iooss & Lemaître, 2015) are also used for global sensitivity analysis and interaction studies, but they decompose output variance and are designed for input-output attribution rather than shaping internal model reasoning. In contrast, we adopt multi-order Shapley interaction (Zhang et al., 2020b), which systematically quantifies feature-pair interactions across contexts and enables direct control of interaction structure during training—providing a model-level summary of interaction complexity beyond what Sobol indices capture. To the best of our knowledge, we are the first to use multi-order Shapley interactions to regulate the fidelity of structured explanations.

## 3 PRELIMINARIES

This section reviews Shapley values and their ability to quantify feature interactions of a model.

Shapley values (Shapley, 1953) provide a principled way to assign credit to input features by averaging their marginal contributions to the model's output across all feature subsets. Let $f(S) : 2^{|N|} \to \mathbb{R}$ be a model that takes a subset of input features $S \subseteq N$ and outputs a scalar prediction, where $N$ is the set of all input features. The remaining features in $N \setminus S$ are replaced with baseline values (e.g., the mean feature values across all samples) (Ancona et al., 2019a). The Shapley value of feature $i$, $\phi(i)$, is computed as the expected marginal contribution of $i$ over all possible feature orderings:

$$\phi(i) = \sum_{S \subseteq N \setminus \{i\}} \frac{|S|!(|N| - |S| - 1)!}{|N|!} \left[ f(S \cup \{i\}) - f(S) \right]. \tag{1}$$

$\phi(i)$ satisfies four desirable properties—*efficiency*, *symmetry*, *dummy*, and *linearity* (Zhang et al., 2020a)—that guarantee fairness in credit allocation to input features. In image classification, each image patch can be treated as an input feature, allowing Shapley values to quantify their individual contributions. While useful for identifying salient image regions, $\phi(i)$ does not capture cooperative effects between image patches. This limitation motivates the formulation of multi-order interaction strength, discussed next.

Shapley values of feature pairs, $\phi(i, j)$, quantify how two features, $(i, j) \in N$, jointly contribute to model predictions across varying input subsets (Zhang et al., 2020a;b). They can be computed at a given $k$th-order of feature interactions, $\phi^{(k)}(i, j)$, to quantify the expected interaction utility between $(i, j) \in N$ in the context of $|S| = k(|N| - 2)$ other input features, $0 \leq k \leq 1$, as

$$\phi^{(k)}(i, j) = \mathbb{E}_{\substack{S \subseteq N \setminus \{i, j\} \\ |S| = k(|N| - 2)}} \left[ f(S \cup \{i, j\}) - f(S \cup \{i\}) - f(S \cup \{j\}) + f(S) \right], \quad 0 \leq k \leq 1. \tag{2}$$

Lower values of $k$ correspond to low-order feature interactions, while higher values capture more complex dependencies. Thus, by varying $k$, $\phi^{(k)}(i, j)$ provides a principled way to probe model reasoning at different complexity levels of feature interactions.

**Multi-order interaction strength.** Following Deng et al. (2022), to quantify a model's feature interactions, we aggregate Shapley-based pairwise interactions into an order-wise summary:

$$J^{(k)} = \frac{\mathbb{E}_{N \in \Omega}[\mathbb{E}_{i, j \in N}[|\phi^{(k)}(i, j)|]]}{\mathbb{E}_{k'}[\mathbb{E}_{N \in \Omega}[\mathbb{E}_{i, j \in N}[|\phi^{(k')}(i, j)|]]]}, \quad 0 \leq k \leq 1, \tag{3}$$

where $\Omega$ is the set of images, each represented by a feature set $N$.

## 4 LINKING INTERACTION STRENGTH TO STRUCTURED EXPLANATIONS

This section begins by reviewing recent work that quantifies structured explanation scale using valid subexplanation counts (Jiang et al., 2024). We then propose a more principled quantification via multi-order Shapley interactions, enabling us to show that structured explanation scale is largely determined by feature-interaction strength.

### 4.1 STRUCTURED EXPLANATIONS AND VALID SUBEXPLANATION COUNTING

Structured Attention Graph (SAG) (Shitole et al., 2021) addresses the limitation of single-explanation attention mechanisms by generating multiple interrelated attention maps that form a graph structure. Each node in the graph represents an attended region (e.g., object parts or contextual features), while edges model their spatial or semantic relationships (e.g., proximity, dependencies). SAG captures complex interactions among diverse image components, offering a holistic understanding of how combinations of features collectively drive predictions.

The scale of SAG was quantified by Jiang et al. (2024) through Minimal Sufficient Explanations (MSE), defined as the smallest subset $S$ of image patches that preserves at least 90% of the model's performance relative to the full image with $N$ patches. A subexplanation is any subset of $S$. The SAG construction begins with the full set $S$ as the root, followed by iteratively removing patches

from $S$ and thereby generating *valid subexplanations* while ensuring the model's softmax confidence score remains above 50% of the original score obtained on $S$.

The number of valid subexplanations in SAG, serves as a metric to characterize the model's reasoning behavior and elucidate fundamental differences in the decision-making mechanisms across various models (Jiang et al., 2024). Models with *disjunctive* reasoning yield smaller subexplanation counts, as they depend on a few critical regions whose removal from $S$ sharply reduces confidence. Conversely, models with more *compositional* reasoning achieve larger counts, since their confidence is distributed across multiple regions and remains stable under partial removal, allowing them to retain high confidence on various subsets of $S$. As demonstrated in Jiang et al. (2024), Transformers exhibit *compositional* behavior, whereas CNNs are characterized by *disjunctive* behavior.

## 4.2 QUANTIFYING EXPLANATION SCALE VIA MULTI-ORDER INTERACTION STRENGTH

Extending (Jiang et al., 2024), we offer a more principled framework which reveals that the scale of structured explanations is determined by multi-order Shapley interaction strength, given by equation 3. This characterization is supported by a correlation analysis between feature-interaction strength and explanation scale, as quantified using subexplanations, on ImageNet (Deng et al., 2009).

**Implementation Details.** Following Deng et al. (2022); Jiang et al. (2024), each image is divided into $16 \times 16$ patches, which define the input feature set $N$, and Shapley interaction strength $J^{(k)}$ is efficiently estimated using the sampling method from Zhang et al. (2020b). Since Jiang et al. (2024) used the first 5,000 images from the ImageNet validation dataset to calculate subexplanation counts, we also use the same dataset to evaluate $J^{(k)}$. For efficiency, 250 patch pairs $(i, j)$ are randomly sampled per image, with $i$ and $j$ no more than two patches apart, since deep models tend to capture stronger interactions between nearby patches. Then, for every $(i, j)$ and every order $k$, we randomly select 250 feature contexts $S \subset N \setminus \{i, j\}$, where $|S| = k(|N| - 2)$, $0 \leq k \leq 1$. These choices are empirically optimized, as detailed in the Appendix A.1. For computing $J^{(k)}$ in equation 3, the model output is defined as $f(S) = \log \frac{P(\hat{y}=c|S)}{1-P(\hat{y}=c|S)}$, where $S$ denotes the selected set of image patches that remain unmasked, while the rest of patches in $N \setminus S$ are masked to average feature values; $P(\hat{y} = c|S)$ is the model's softmax score for class $c$ given the visible subset $S$; and $c = \arg\max_{\hat{y}} P(\hat{y}|N)$ is the class that the model predicts for the full input $N$. $J^{(k)}$ is evaluated for various models covering multiple design paradigms, as in Jiang et al. (2024), including: older CNNs—VGG19 (Simonyan & Zisserman, 2015), ResNet50 (He et al., 2016); newer CNNs—ResNet50-C1, ResNet50-C2, ResNet50-D (Wightman et al., 2021); hybrid convolution-transformer models—ConvNeXt-T (Liu et al., 2022); Transformers—Swin-T (Liu et al., 2021), Nest-T (Zhang et al., 2022), DeiT-S (Touvron et al., 2021), PiT-S (Heo et al., 2021)); and distilled Transformer variants—DeiT-S-distilled (Touvron et al., 2021), PiT-S-distilled (Heo et al., 2021), LeViT-256 (Graham et al., 2021). All pretrained models were sourced from the torchvision and timm libraries. Experiments were carried out using 2 Nvidia H100 80GB GPUs or 4 Nvidia Tesla V100 32GB GPUs. We report the time and computational cost of interaction strength estimation in Appendix A.2.

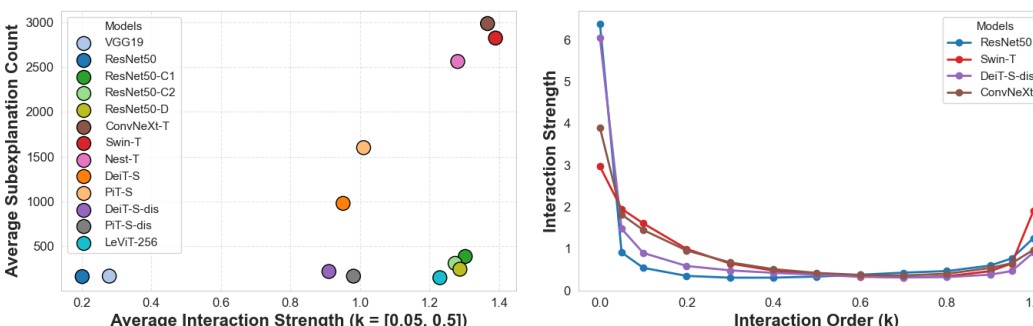

Figure 1: Subexplanation count vs interaction strength for orders $k \in [0.05, 0.5]$ on ImageNet. Similar architectures cluster together, suggesting a positive correlation between the two metrics.

Figure 2: Multi-order interaction strength across four models trained on ImageNet. Transformers capture stronger feature interactions across intermediate spatial contexts $0.1 < k < 0.4$.

Fig. 1 shows the correlation between $J^{(k)}$, given by equation 3, and the subexplanations counts reported in Jiang et al. (2024), across diverse models trained on ImageNet, for intermediate spatial contexts $S$ taking up to a 50% of the input image, $k \in [0.05, 0.5]$. As shown, models with similar architectures tend to cluster together along the two explanation metrics. Transformers, such as Swin-T and Nest-T, exhibit both lager subexpalnation counts and high interaction strengths, whereas CNNs, such as VGG19 and ResNet50, are characterized by smaller subexpalnation counts and low interaction strengths. This suggests a strong positive correlation between the two metrics and indicates that explanation scale is tightly linked to the types of feature interactions a model encodes. To further support this trend, we computed correlation values (Essam et al., 2022) across all 13 models in Fig. 1, showing a moderate linear correlation (Pearson = 0.47) and strong monotonic trends (Spearman = 0.66, Kendall = 0.51), confirming that models with higher interaction strength consistently exhibit larger subexplanation counts.

Fig. 2 further illustrates our characterization of structured explanation scales for four representative models in terms of $J^{(k)}$. As shown, CNNs on ImageNet exhibit stronger low-order interactions over small spatial contexts $0 < k < 0.1$, with notably weaker interactions at mid $k$. This pattern echoes the "representation bottleneck" discussed by Deng et al. (2022), where CNNs generally struggle to encode interactions of moderate complexity. Our results provide further insight that this phenomenon is architecture-dependent — while CNNs strongly exhibit the bottleneck, Transformers capture stronger interactions across intermediate spatial contexts $(0.1 < k < 0.4)$. This is intuitive, as the self-attention mechanism enables modeling of broader spatial relationships. Notably, this distinction aligns with subexplanation counts (higher for Transformers, lower for CNNs), reinforcing the view that interaction structure provides a principled lens into how different models reason. A comprehensive set of per-model heatmaps—including CNNs, hybrids, and Transformers—is provided in Appendix A.3 to illustrate these patterns in greater detail.

## 5 CONTROLLING STRUCTURED EXPLANATIONS BY INTERACTION LOSS

In this section, we build on the empirical observation that feature-interaction strength correlates with structured explanation scale. Motivated by this insight, we investigate whether incorporating XAI objectives into training via interaction-based regularization could steer structured explanation scales toward a desired level and influence model reasoning behavior. We first outline the training strategy that explicitly controls interaction strength. Then we examine how explicit training for feature interactions of specific complexity alters the model's reasoning behavior, as interpreted through two explainability frameworks: SAG (Shitole et al., 2021) and ProtoTree (Nauta et al., 2021).

### 5.1 SHAPLEY INTERACTION-BASED LOSS FUNCTIONS

Following Deng et al. (2022), we adopt two loss functions based on multi-order interaction strength to explicitly train the model to favor feature interactions of a given order. The two losses are specified in terms of a prediction-change distribution over $C$ classes, $U(k_1, k_2) = \{U_c(k_1, k_2) : c = 1, \ldots, C\}$, which quantifies how predictions change as the amount of unmasked spatial context at the input varies using

$$U_c(k_1, k_2) = \text{softmax}\left(\mathbb{E}_{S_1 \subset S_2 \subseteq N}[z_c(S_2) - \frac{k_2}{k_1}z_c(S_1)]\right), \tag{4}$$

where $S_1$ and $S_2$ are random subsets of input features, $|S_1|=k_1|N|, |S_2|=k_2|N|, 0<k_1<k_2\leq 1$, and $z_c(S)$ is the model's logit for class $c$ when features of image patches in $N \setminus S$ are masked to their dataset-wide average values. Larger values of $U_c(k_1, k_2)$ indicate greater changes in the prediction across the range of interaction orders $(k_1, k_2)$. The two loss functions are defined as:

$$L^+(k_1, k_2) = -\sum_{c=1}^{C} P(y^* = c) \log U_c(k_1, k_2), \quad L^-(k_1, k_2) = \sum_{c=1}^{C} U_c(k_1, k_2) \log U_c(k_1, k_2), \tag{5}$$

where $P(y^* = c)$ is the true probability of class $c$ for the input image (e.g., 1 for the ground-truth class). Alongside the standard classification loss, we use these two additional loss functions to train the model with the following total loss: $L = L_{\text{classification}} + \lambda L^+(k_1, k_2) + (1 - \lambda)L^-(k_1, k_2)$, where $\lambda = \{0, 1\}$ controls which loss is active during training. Minimizing $L^+$ encourages the

model to be more discriminative across the given interaction orders $[k_1, k_2]$, thereby increasing the strength of feature interactions within this range. Since $L^-$ is the negative entropy of the distribution $U(k_1, k_2)$, minimizing $L^-$ promotes non-discriminative behavior across the specified interval $[k_1, k_2]$, effectively suppressing interaction strength in this range. Importantly, the training loss does not require computing Shapley values or interaction strengths during optimization. It only involves an additional forward–backward pass on two masked variants of the input, adding a modest, linear overhead per epoch that remains practical even on standard hardware.

## 5.2 EFFECTS ON STRUCTURED ATTENTION GRAPH

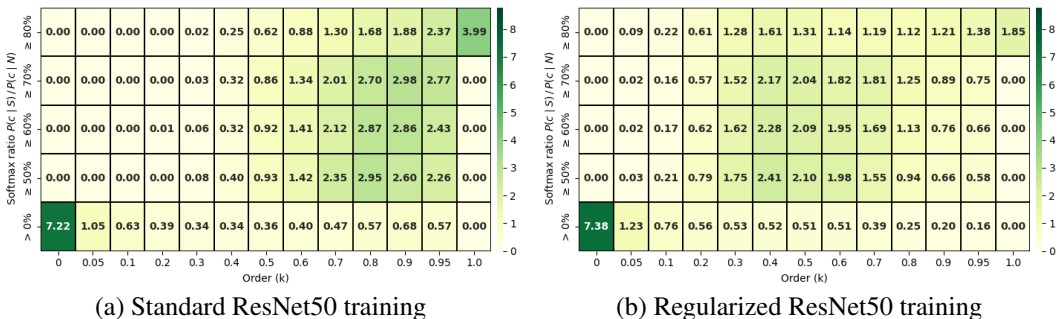

(a) Standard ResNet50 training    (b) Regularized ResNet50 training

Figure 3: Heatmaps of $J^{(k)}$ across varying softmax confidence ratios $\frac{P(c|S)}{P(c|N)}$ and orders $k$ for ResNet50 on ImageNet: (Left) standard training, (Right) training with $L^-(0.7, 1)$ regularization enhances mid-order interactions compared to standard training.

**Impact of $L^-$ Regularization on CNNs.** Inspired by the reasoning behavior of Transformers—strong feature interactions at intermediate orders $k$ shown in Fig. 1, 2—we aim to shift the reasoning of ResNet50 toward more Transformer-like, compositional behavior. We expect that promoting mid-order interactions during training will encourage richer integration of visual cues across broader spatial contexts, supporting more interpretable and robust reasoning under partial occlusions and other visual challenges. To this end, we apply $L^-(0.7, 1)$ in the total loss ($\lambda = 0$) during training to suppress higher-order and encourage mid-order feature interactions. To summarize interaction strength, we average $J^{(k)}$ within five bins defined by $\frac{P(c|S)}{P(c|N)} \geq b$, with $b \in \{80\%, 70\%, 60\%, 50\%, 0\%\}$, following the subexplanation thresholds in Jiang et al. (2024). As shown in Fig. 3, the large values of interaction strength, previously concentrated at higher $k$ under standard training (Fig. 3a), shift toward mid-order interactions (Fig. 3b), as intended.

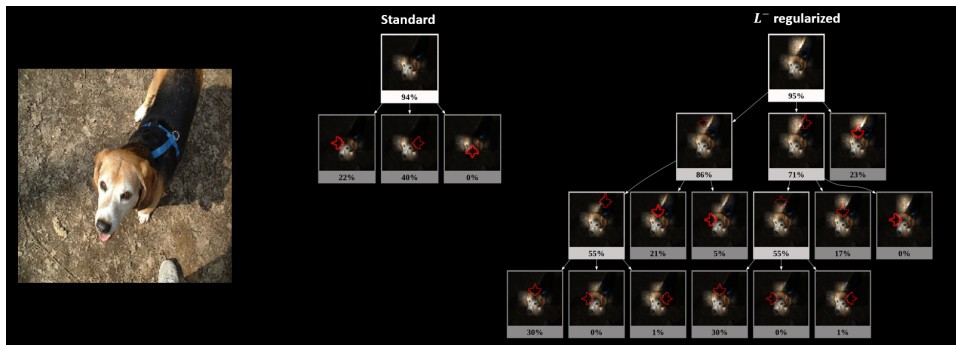

Figure 4: SAG explanations of ResNet50 regularized with $L^-(0.7, 1)$. The red regions denote the patches removed at each SAG node, and the confidence shown reflects the model's prediction on the masked image. Our regularized training shifts the model's focus from localized head regions (left) to holistic body regions (right).

We next examine how regularized training alters the organization of explanations in SAG. For ResNet50, training with $L^-(0.7, 1)$ produces longer explanation paths in SAG, indicating that the model distributes attribution across a broader set of regions rather than concentrated in a few patches

Table 1: Subexplanation counts of ResNet50 and SWIN-T on ImageNet

| Model | Training type | Subexplanation counts across softmax confidence ratios | | | |
| | | $\geq 80\%$ | $\geq 70\%$ | $\geq 60\%$ | $\geq 50\%$ |
|---|---|---|---|---|---|
| ResNet50 | Standard | 53.7 | 108.6 | 180.4 | 296.9 |
| | $L^-(0.7, 1)$ reg. | 86.9 | 168.8 | 264.4 | 378.4 |
| SWIN-T | Standard | 221.58 | 882.72 | 2933.03 | 7268.20 |
| | $L^+(0, 0.5)$ reg. | 55.39 | 122.70 | 290.97 | 636.41 |

(Fig. 4). This change is quantified through subexplanation counts (Jiang et al., 2024), and as further evidence of the positive correlation between subexplanation counts and Shapley interaction strength, Tab. 1 reports that after training with $L^-(0.7, 1)$ regularization, ResNet50's subexplanation counts increase across varying softmax confidence ratios $\frac{P(c|S)}{P(c|N)}$ on ImageNet (Deng et al., 2009). Classification performance remains comparable. The higher subexplanation counts indicate that we encouraged ResNet50 to rely on a broader yet more focused set of patches—shifting from diffuse high-order to compositional mid-order reasoning, as hypothesized. We conducted the same analysis for fine-grained classification on the CUB-200-2011 dataset (Wah et al., 2011) and observed a similar trend, with values reported in Appendix A.4.

Table 2: Accuracy (%) under salient region removal. Values show top-1 accuracy as increasing percentages of the most salient pixels are removed (SmoothGrad).

| Training | 0% | 10% | 20% | 30% | 40% | 50% | 60% | 70% | 80% | 90% |
|---|---|---|---|---|---|---|---|---|---|---|
| Standard | 99.29 | 67.66 | 45.26 | 32.59 | 24.66 | 20.25 | 18.59 | 18.08 | 16.22 | 13.22 |
| $L^-(0.7, 1)$ | 99.08 | 96.96 | 92.56 | 82.20 | 66.29 | 46.93 | 28.51 | 17.38 | 12.55 | 12.12 |
| $L^-(0.1, 1)$ | 99.00 | 98.02 | 95.93 | 93.23 | 88.44 | 81.52 | 67.9 | 46.85 | 27.45 | 15.71 |

Finally, we contrast our method with prior regularization-based approaches. Unlike methods that focus on a single isolated attribution (e.g., saliency alignment in Ismail et al. (2021)), our method shapes *structured* explanations by modulating feature interactions. To illustrate this distinction, Tab. 2 presents the saliency-removal evaluation from Ismail et al. (2021). We trained their CNN with and without the $L^-$ loss (for two configurations of $k_1$ and $k_2$) and measured the model accuracy as we incrementally removed the most salient pixels identified by SmoothGrad. As shown, the CNN trained without $L^-$ suffers a sharp accuracy drop — indicating overreliance on a few highly salient regions — whereas the model trained with $L^-$ degrades more gradually, suggesting that predictions depend on broader, more distributed feature interactions (as explicitly intended during training).

Importantly, the CNN trained with $L^-(0.1, 1.0)$ exhibits a slower accuracy decline than the model trained with $L^-(0.7, 1.0)$ as salient regions are progressively removed. This indicates that $L^-(0.7, 1.0)$ promotes broader but more distributed dependencies, while $L^-(0.1, 1.0)$ encourages reliance on many smaller, independent patch subsets that remain robust even under substantial deletion. These results highlight our method's fine-grained control over the structure of model explanations—not by steering attention to specific regions, but by shaping how explanatory content is distributed across interactions. This supports our central claim: rather than competing with saliency-based or "right-reason" training methods, our approach complements them by offering explicit control over the form and granularity of explanations.

**Impact of $L^+$ Regularization on Transformers.** Given that Transformers exhibit strong mid-order feature interactions, we apply $L^+(0, 0.5)$ regularization ($\lambda = 1$) during Swin-T training to boost lower-order $J^{(k)}$ values, thereby steering the model's reasoning toward a more CNN-like, disjunctive behavior—if such behavior is desired. A comparison between interaction strengths after standard training and after training with $L^+$ regularization on ImageNet, shown in Fig. 5, demonstrates that low-order interaction strengths have increased, as intended. Turning to qualitative explanations, SAG reveals shorter explanation paths under $L^+(0, 0.5)$, where attribution becomes more concentrated and foreground-focused (Fig. 6). These qualitative observations are further supported

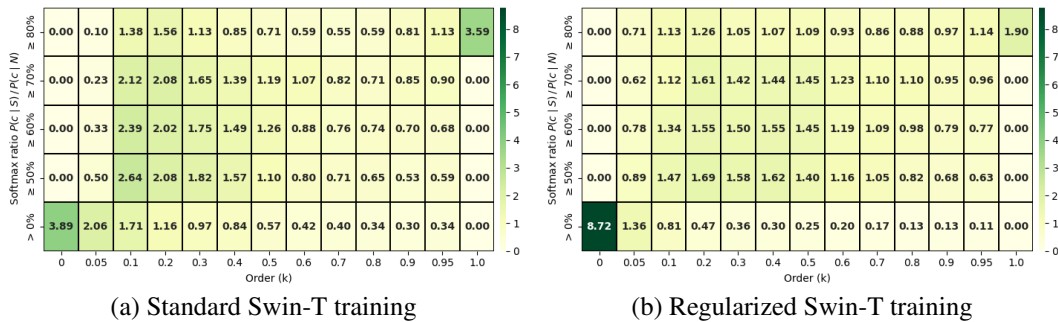

(a) Standard Swin-T training          (b) Regularized Swin-T training

Figure 5: Heatmaps of $J^{(k)}$ across varying softmax confidence ratios $\frac{P(c|S)}{P(c|N)}$ and orders $k$ for Swin-T on ImageNet: (Left) standard training, (Right) training with $L^+(0, 0.5)$ regularization enhances lower-order interactions compared to standard training.

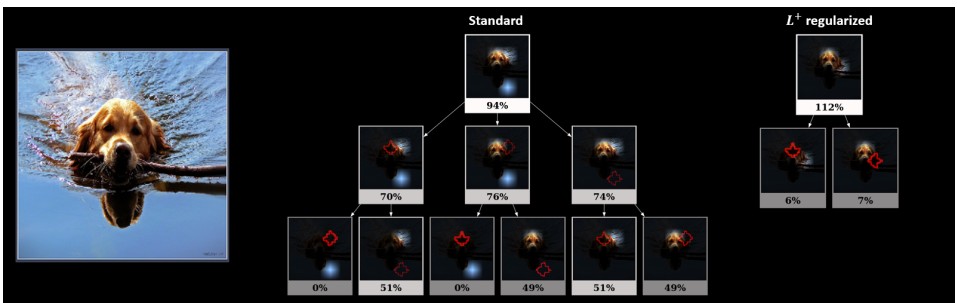

Figure 6: Effect of $L^+(0, 0.5)$ on Swin-T. The red regions indicate the patches removed at each SAG node, and the confidence corresponds to the model's response after this removal. Our regularized training shifts the model's attention from extraneous background features (left) to semantically relevant foreground regions (right).

by quantitative analysis - Tab. 1 shows that training Swin-T on ImageNet with $L^+(0, 0.5)$ regularization leads to consistently lower subexplanation counts across a range of softmax confidence ratios $\frac{P(c|S)}{P(c|N)}$. This reduction suggests that the applied regularization effectively steers the model toward weaker compositional reasoning by decreasing the scale of its structured explanations. These findings further support a positive correlation between subexplanation counts and $J^{(k)}$.

Overall, these findings demonstrate that interaction-based regularization provides controllability over both the model's reasoning behavior and the form of its explanations, without implying that any particular style is inherently superior or more interpretable.

### 5.3 EFFECTS ON PROTOTREE INTERPRETABILITY

ProtoTree (Wu et al., 2024; Xu-Darme et al., 2023; Rymarczyk et al., 2022; Cui et al., 2023) replaces the model's standard classification layer with a decision tree composed of learned prototypes. Each prototype determines soft routing decisions in the decision tree based on its similarity to image patches, enabling the model to perform classification through a sequence of interpretable binary decisions. As detailed in Nauta et al. (2021), the learned prototypes are visualized via similarity mapping to training patches – ensuring interpretable decision processes.

While ProtoTree offers inherent transparency via its prototype-based reasoning, we observe that many prototypes can still activate on irrelevant or background regions, limiting their semantic value. In this work, we hypothesize that incorporating interaction-based regularization during training encourages prototypes to attend to foreground object regions that are generally more informative for classification. Our aim is to modify interpretability not by post-hoc filtering or visualizations, but by influencing what the model attends to during learning.

To evaluate this hypothesis, we quantify $L^-(0.7, 1)$ regularization's impact on interpretability via prototype-foreground overlap precision. Prototypes overlapping image foregrounds (vs. backgrounds) are prioritized as interpretable, with true positives requiring prototype–ground-truth foreground alignment. The precision is evaluated via the Intersection over Union (IoU) between the prototype region and the relevant annotation, with IoU thresholds from 50% to 75% in 5% steps.

We trained standard and $L^-(0.7, 1)$-regularized ProtoTree variants on ImageNet-200 (200 classes, 14k training and 6k validation images with bounding box annotations). The resulting trees consisted of 203 and 199 prototypes, respectively, and both models achieved comparable classification accuracy (83.12% and 83.17%). We applied the same training setup and regularization strategy for CUB-200-2011. The standard and regularized ProtoTrees trained on CUB yielded similar performance (81.74% and 81.31%), with 202 and 203 final prototypes, respectively. This consistency in classification performance between the standard and regularized models on both datasets ensures that observed interpretability differences are not driven by performance disparities.

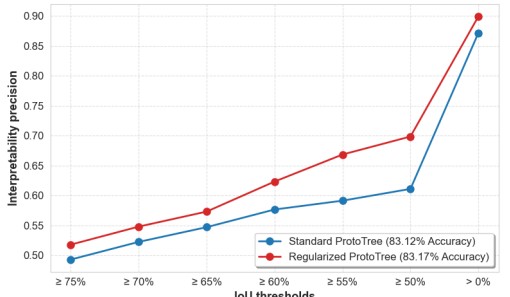 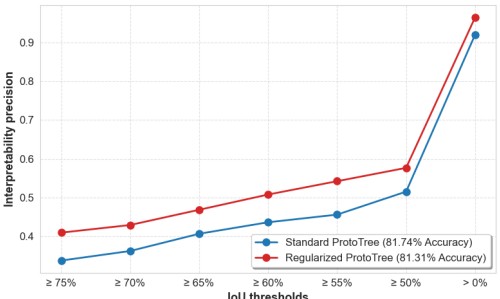

Figure 7: Improved precision of prototypes for our regularized ProtoTree compared to standard ProtoTree on ImageNet-200 (left), CUB-200-2011 (right).

As shown in Fig. 7, the regularized ProtoTree consistently achieved higher interpretability precision on both ImageNet-200 and CUB-200-2011. Compared to the standard model, more prototypes aligned with semantically meaningful object regions and fewer activated on background areas. This supports our hypothesis that interaction-based regularization guides the model toward more semantically meaningful representations and encourages ProtoTree to form more interpretable explanations. A complementary part-level analysis on CUB-200-2011 (Appendix A.5) further evaluates prototype alignment with semantic parts (e.g., beak, wing, tail), providing finer granularity. Qualitative examples comparing prototype-to-region alignments are provided in Appendix A.6, illustrating the same trend. All evaluations are grounded in annotation-based alignment, avoiding reliance on subjective human judgments. Together, these results demonstrate that interaction-based regularization improves both the precision and structural grounding of ProtoTree explanations.

## 6 CONCLUSION

To bridge network training with post hoc interpretability, we adopt a game-theoretic framework based on Shapley values and show that incorporating Shapley interaction strength into the training objective enables direct control over the scale and fidelity of structured explanations. Networks trained with this regularization produce structured explanations—such as SAG and ProtoTrees—that reflect shifts in reasoning between holistic feature integration and sparse, localized cues. These changes are quantified by linking the number of Minimal Sufficient Subexplanations to Shapley interaction strength. The results show that the typically disjunctive reasoning of CNNs can be steered toward compositional, while the compositional reasoning of Transformers can be made more disjunctive. We do not assume that any particular interaction order is inherently superior; rather, the value of controllability lies in adapting explanation structure to deployment needs, as prior work shows different structures can enhance robustness, modularity, or usability, and also improve user understanding and counterfactual reasoning. Moreover, we show that the proposed regularization improves the fidelity of structured explanations by aligning them more closely with annotation-grounded object and part regions. This is quantified by evaluating the true-positive overlap between

ProtoTree explanations and ground-truth foreground regions in both large-scale (ImageNet) and fine-grained (CUB-200-2011) classification settings. By characterizing various architectures with multi-order Shapley interaction strength and correlating these with subexplanation counts, we offer a more principled framework for studying the scale of structured explanations. Limitations include the use of approximation-based Shapley estimates and evaluations restricted to specific structured explanation methods. Exploring broader tradeoffs among accuracy, interpretability, robustness, and controllability remains future work.

## 7 REPRODUCIBILITY STATEMENT

This work adopts a Shapley-based interaction metric and loss functions for explanatory regularization. Sec. 3 provides the theoretical formulation, while Sec. 4.2 and Appendices A.1, A.2 describe the estimation of interaction strengths, including implementation details, sampling strategies, and computational costs. Sec. 5.1 specifies the proposed loss functions, while Sec. 5.2, 5.3 detail the training setups for CNNs, Transformers, and ProtoTrees, ensuring reproducibility.

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

## A APPENDIX

### A.1 EFFECT OF SAMPLING PARAMETERS

We conduct an ablation study to analyze how the choice of the number of patch pairs $M$ and number of contexts $C$ affect the approximation of $J^{(k)}$. The study assesses the stability of the approximated $J^{(k)}$ using an instability metric (Deng et al., 2022). Specifically, we calculate $J^{(k)}$ separately for each input image represented by a feature set $N$ as:

$$J^{(k)}(N) = \frac{\mathbb{E}_{i,j \in N}[|\phi^{(k)}(i,j)|]}{\mathbb{E}_{k'}[\mathbb{E}_{i,j \in N}[|\phi^{(k')}(i,j)|]]}, \quad 0 \le k \le 1, \tag{6}$$

The overall instability of $J^{(k)}$ was then computed as:

$$\text{instability} = \mathbb{E}_{N \in \Omega} \mathbb{E}_k \left[ \frac{\mathbb{E}_{u,v;u \neq v}|J_u^{(k)}(N) - J_v^{(k)}(N)|}{\mathbb{E}_w |J_w^{(k)}(N)|} \right]. \tag{7}$$

where $J_u^{(k)}(x)$ and $J_v^{(k)}(x)$ represent the approximated values of $J^{(k)}(N)$ at the $u$-th and $v$-th sampling times, respectively. This instability metric helps quantify the variability in the approximation of $J^{(k)}$ across different sampling iterations and interaction orders.

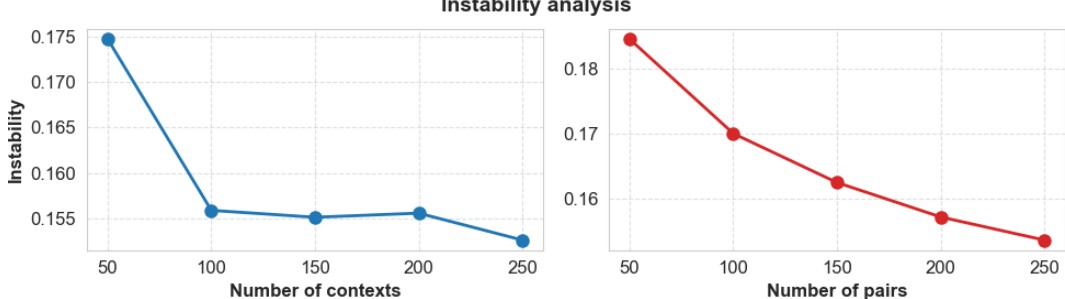

Figure 8: The instability of $J^{(k)}$ $w.r.t$ the number of patch pairs $M$ and number of contexts $C$.

We evaluated the instability using a ResNet50 model trained on the ImageNet dataset. We conducted two experiments to analyze how the number of sampled patch pairs $M$ and number of sampled contexts $C$ affected instability. In the first experiment, we fixed the number of pairs at 100 and examined the impact of varying the number of sampled contexts. Fig. 8 shows as the number of contexts increased, the instability gradually declined. When the number of contexts reached 250, the instability dropped to approximately 0.15, confirming that a sufficient number of sampled contexts improved stability. Based on this result, we fixed the number of contexts at 250 in the second experiment and varied the number of sampled patch pairs. Similarly, Fig. 8 shows that increasing the number of sampled pairs reduces instability. Once the number of pairs surpassed 250, the instability dropped below 0.15, indicating a stable approximation of interaction strength. These findings demonstrate that our sampling strategy provides a reliable approximation of $J^{(k)}$.

Beyond assessing the stability of $J^{(k)}$, we also examine the storage requirements associated with varying the number of patch pairs $M$ and number of contexts $C$. Specifically, we track the disk space used to store intermediate logits, which are saved as *.npy* files to prevent exceeding GPU memory constraints. The evaluation follows a similar experimental setup as in the instability analysis. Fig. 9 shows that in the first experiment with 100 fixed patch pairs, as the number of contexts increases beyond 250, storage requirements exceed 250GB. Likewise, in the second experiment with a fixed number of 250 contexts, when the number of patch-pairs surpasses 250, storage usage exceeds 640GB. These findings highlight the substantial storage overhead introduced by larger sampling choices, underscoring the trade-off between approximation stability and storage feasibility.

**Selection of Optimal Parameters.** Considering the trade-off between stability and storage constraints, we select the number of $(i, j)$ pairs, $M = 250$ and number of contexts, $C = 250$ as the

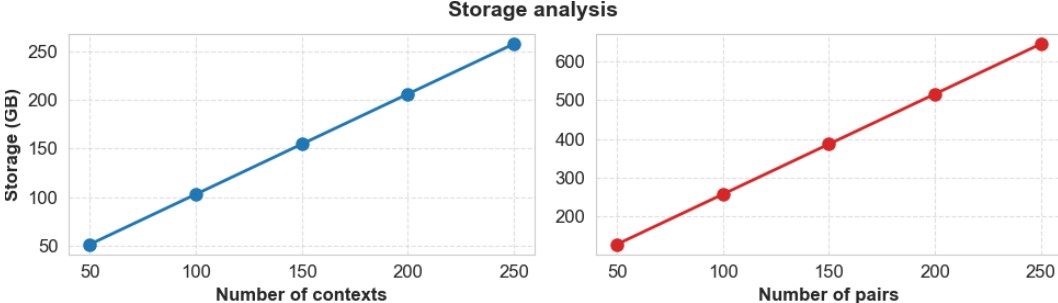

Figure 9: The required storage for approximating $J^{(k)}$ $w.r.t$ the number of patch pairs $M$ and number of contexts $C$.

optimal values for our evaluations. These settings ensure a stable approximation of $J^{(k)}$ while keeping storage requirements within a feasible range. Choosing smaller values risks increased instability, whereas larger values lead to excessive storage consumption without significant gains in stability. By adopting these optimal parameters, we strike a balance between computational efficiency and reliable interaction strength estimation.

## A.2 INTERACTION STRENGTH ESTIMATION COMPLEXITY

We analyze the time and computational cost required to estimate interaction strengths for a single image. The total cost scales with the number of patch pairs $M$ and the number of context sets $C$ sampled per pair. For each $(i, j)$ pair, $C$ masked context variants are generated, and four forward passes are required per context to compute $[f(S \cup \{i, j\}) - f(S \cup \{i\}) - f(S \cup \{j\}) + f(S)]$. Tab. 3 summarizes empirical runtimes and average FLOPs for two representative models—ResNet50 and Swin-T—using 2 Nvidia H100 80GB GPUs.

Table 3: Interaction computation time and FLOP cost for one image. $M$: number of patch pairs; $C$: number of context samples per pair.

| Model | GFLOPs | $M$ | $C$ | Time (s) |
|-------|--------|-----|-----|----------|
| ResNet50 | 4.13 | 250 | 250 | 348.12 |
|          |      |     | 100 | 141.76 |
| Swin-T | 4.37 | 250 | 250 | 694.60 |
|        |      |     | 100 | 279.32 |

Tab. 3 shows that ResNet50 requires 4.13 GFLOPs per forward pass. For $M = 250$ and $C = 250$, computing interactions takes 348.12 seconds, in addition to 13.17 seconds for sampling contexts. Reducing the number of contexts to $C = 100$ lowers the interaction computation time to 141.76 seconds and sampling time to 5.32 seconds. Swin-T requires slightly higher cost per forward pass (4.37 GFLOPs). For the same setting with $M = 250$ and $C = 250$, interaction computation takes 694.60 seconds, while reducing to $C = 100$ lowers it to 279.32 seconds. Context sampling overheads are identical to those of ResNet50. These results confirm that total cost grows linearly with both the number of context samples and patch pairs, and that Swin-T incurs higher wall-clock time due to its greater computational complexity per inference.

The computational cost of our framework can be flexibly controlled by adjusting the number of patch pairs $M$ and sampled contexts per pair $C$, allowing practitioners to trade off interpretability resolution against runtime. For each image, the method requires $(4 \times M \times C)$ forward passes, resulting in a total cost of approximately $4MC \times$ FLOPs, where the per-pass FLOPs depend on the model architecture (e.g., 4.13 GFLOPs for ResNet50 and 4.37 GFLOPs for Swin-T). This linear scaling makes the method tunable for different hardware budgets, and practical even on moderately resourced systems. As expected, total runtime is further influenced by model-specific inference complexity and GPU throughput, with larger architectures such as Swin-T exhibiting longer per-image computation times.

A.3   ADDITIONAL EVALUATION OF INTERACTION STRENGTHS ACROSS MODELS

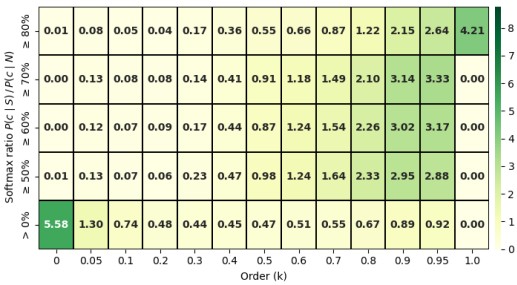

Figure 10: Multi-order interaction strengths for VGG19 trained on ImageNet.

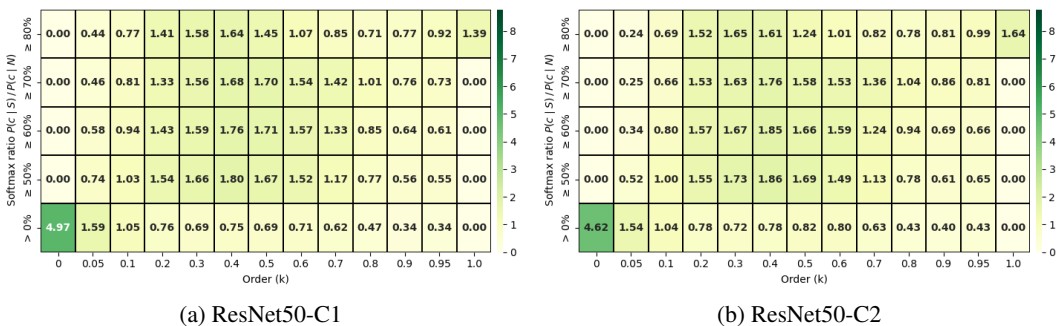

(a) ResNet50-C1          (b) ResNet50-C2

Figure 11: Multi-order interaction strengths for ResNet50-C1 and ResNet50-C2 trained on ImagNet.

**ImageNet.** Fig. 10 shows the interaction heatmap for VGG19. Like ResNet50, VGG19 exhibits a distinctly bimodal structure, with the highest interaction strength concentrated at $k = 0$ under low-confidence predictions (5.58 when likelihood ratio $< 50\%$) and a secondary rise at high orders ($k \geq 0.8$), peaking at 4.21. However, interaction strengths across the middle-order range remain minimal, reinforcing the presence of a representational bottleneck. This pattern highlights VGG19's strong reliance on either highly localized or fully global feature combinations, and its limited ability to model interactions involving a moderate number of contextual variables.

Fig. 11 and Fig. 12 extend the interaction strength analysis to additional CNN variants and hybrid architectures. ResNet50-C1 and ResNet50-C2 (Fig. 11) continue to exhibit the characteristic low-order peak seen in standard CNNs, with strong interaction strength at $k = 0$ for low-confidence predictions (4.97 for C1 and 4.62 for C2 when likelihood ratio $< 50\%$). However, unlike VGG19 and standard ResNet50, the interaction distribution for C1 and C2 is less sharply bimodal. Specifically, interaction strength at high orders ($k \geq 0.9$) is notably reduced, and mid-level orders (e.g., $k \in [0.2, 0.6]$) show modest increases in strength, suggesting a mild shift toward more distributed reasoning. This hints at a partial relaxation of the representational bottleneck, though the core CNN limitation of reduced capacity for encoding interactions at intermediate orders remains evident.

ResNet50-D and ConvNeXt-T (Fig. 12) continue this progression. Like ResNet50-C1 and C2, ResNet50-D shows a strong interaction peak at $k = 0$ under low-confidence conditions (3.30 for likelihood ratio $< 50\%$), but the drop across middle orders is less severe, and high-order interactions are slightly diminished. This reflects a gradual shift toward more distributed interactions without fundamentally departing from the representational limitations of conventional CNNs.

ConvNeXt-T, on the other hand, demonstrates a noticeably different interaction profile. While it retains a low-order peak (3.75 at $k = 0$ for low-confidence inputs), it sustains stronger interaction strengths across the middle-order range ($k \in [0.1, 0.6]$) and shows only a mild rise at high orders. This distribution indicates a more balanced encoding of interactions which are less dependent on isolated or extreme contexts, suggesting improved capacity for mid-level feature integration. These trends are consistent with ConvNeXt's architecture, which replaces ReLU with GELU, BatchNorm

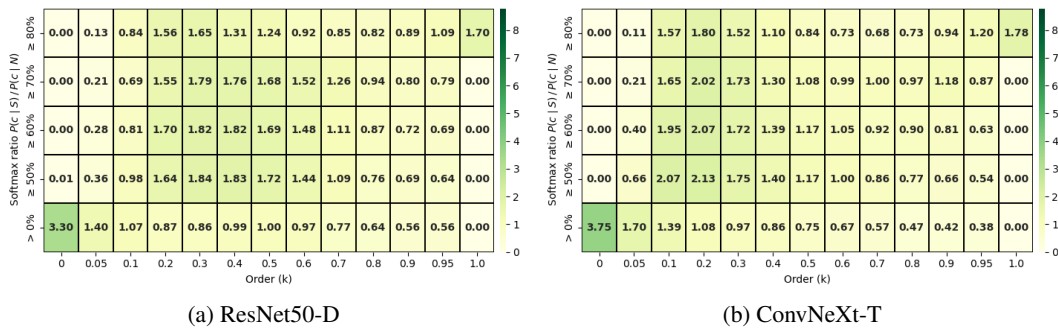

(a) ResNet50-D  (b) ConvNeXt-T

Figure 12: Multi-order interaction strengths for ResNet50-D and ConvNeXt-T trained on ImagNet.

with LayerNorm, and introduces depthwise convolutions with large kernel sizes—features adapted from Transformer models that likely contribute to its more compositional reasoning behavior.

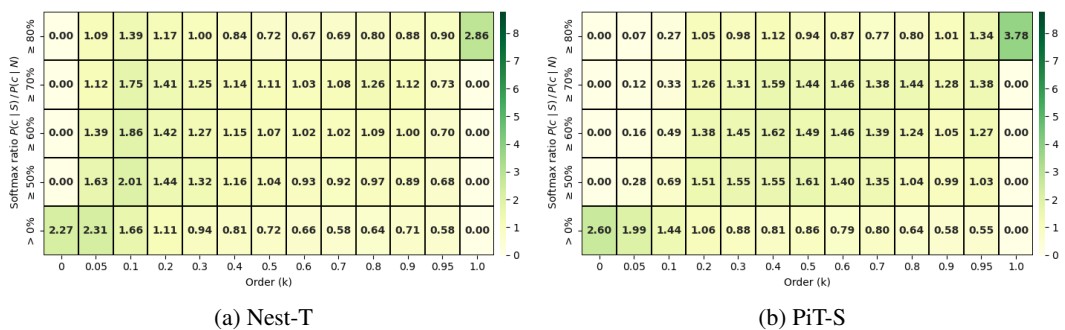

(a) Nest-T  (b) PiT-S

Figure 13: Multi-order interaction strengths for Nest-T and PiT-S trained on ImagNet.

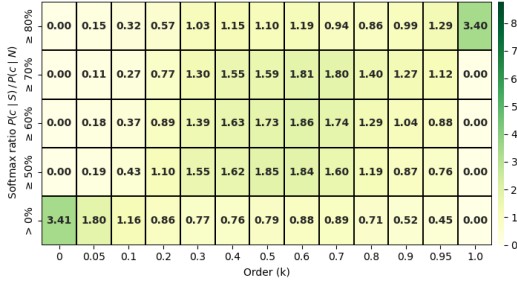

Figure 14: Multi-order interaction strengths for DeiT-S trained on ImagNet.

Fig. 13 presents interaction heatmaps for Nest-T and PiT-S, which follow the typical behavior seen in Transformer-based models such as Swin-T and DeiT-S. Both exhibit a more balanced interaction distribution compared to CNNs, with meaningful interaction strength maintained across intermediate orders ($k \in [0.1, 0.6]$), even at higher confidence levels. For instance, Nest-T peaks modestly at $k = 0$ under low-confidence conditions (2.27), and sustains elevated interaction strengths through middle orders as confidence increases. PiT-S shows a similar pattern with a relatively mild peak at order 0 (2.60), and strong contributions at high orders (e.g., 3.78 at $k = 1.0$), underscoring its ability to engage distributed features across the spatial extent of the image. These trends reinforce the earlier observation that self-attention architectures favor compositional and context-aware representations, distributing interactions more evenly rather than relying on extremes.

Fig. 14 shows the interaction heatmap for DeiT-S. Consistent with other Transformer-based models, DeiT-S displays a more distributed interaction pattern, with significant strength across a broad span of intermediate orders ($k \in [0.2, 0.7]$). While the interaction strength still peaks at $k = 0$ under

low-confidence predictions (3.41), the increase is less pronounced than in CNNs, and middle-order contributions remain substantial across all confidence tiers. This distribution supports the model's capacity for context-dependent reasoning and confirms that DeiT-S maintains the Transformer trait of avoiding sharp reliance on low or high extremes—favoring instead a smoother, compositional integration of features across the input.

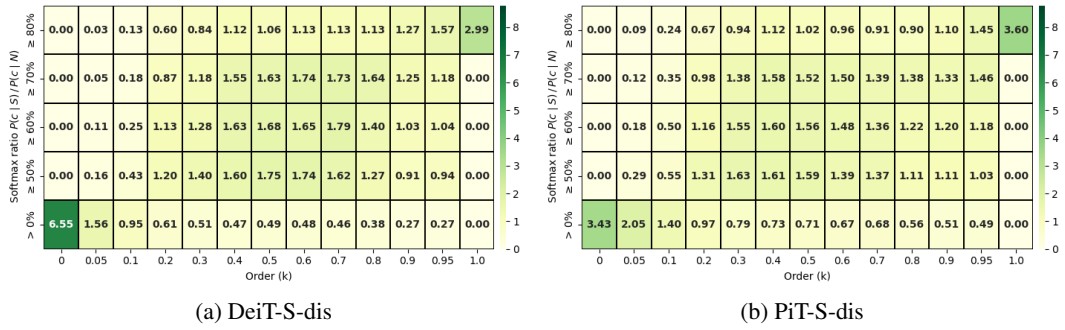

(a) DeiT-S-dis               (b) PiT-S-dis

Figure 15: Multi-order interaction strengths for DeiT-S-dis and PiT-S-dis trained on ImagNet.

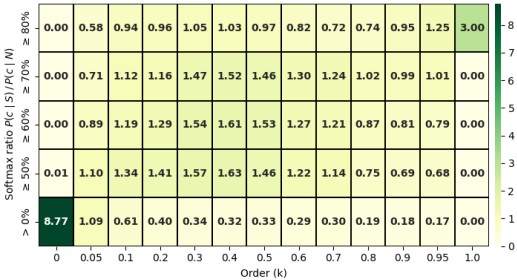

Figure 16: Multi-order interaction strengths for LeViT-256 trained on ImagNet.

In contrast, the interaction profiles of the distilled Transformer variants (Fig. 15 and Fig. 16) resemble those of classical CNNs. DeiT-S-dis exhibits a sharp spike at order 0 under low-confidence settings (6.55), with significantly diminished strength across both mid and high orders. Similarly, PiT-S-dis peaks at 3.43 for $k = 0$, followed by a steep decline across the context spectrum. LeViT-256 further amplifies this trend, showing an extreme low-order peak of 8.77, surpassing even the highest values seen in ResNet50 and VGG19, while mid and high-order interactions are nearly absent. This shift mirrors the dual-peak structure noted in CNNs and suggests that distillation, while effective for compressing or simplifying models, may inadvertently reintroduce a representational bottleneck. The return to low-order dominance highlights a potential trade-off introduced by distillation: it may aid efficiency and training stability, but appears to constrain the model's capacity for compositional, multi-part reasoning.

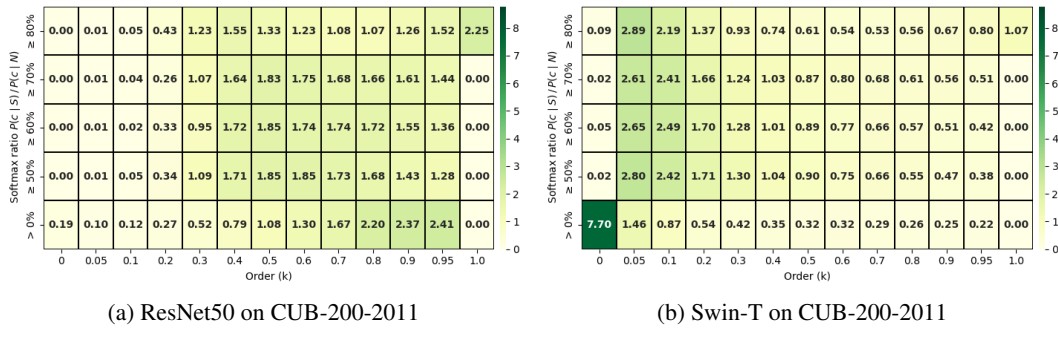

(a) ResNet50 on CUB-200-2011        (b) Swin-T on CUB-200-2011

Figure 17: Multi-order interaction strengths for ResNet50 and Swin-T trained on CUB-200-2011.

**CUB-200-2011.** Fig. 17a shows that ResNet50 on CUB-200-2011 exhibits significantly higher $J^{(k)}$ values, across a wide range of mid to high orders $k$, compared to ImageNet, suggesting a more distributed reasoning process involving multiple features. This behavior likely reflects the increased modeling demands of the fine-grained recognition task. On the other hand, Fig. 17b shows that Swin-T on CUB-200-2011 exhibits a shift in its multi-order interaction strength toward order 0 under low-confidence settings. This suggests that, when uncertain, the Transformer defaults to localized low-order spatial contexts rather than engaging in compositional reasoning. These results highlight that the proposed multi-order interaction strength effectively captures how the task domain—fine-grained versus general classification—influences the reasoning mechanisms of different architectures.

## A.4 ADDITIONAL REGULARIZATION ON CUB-200-2011

Table 4: Subexplanation counts of ResNet50 on CUB-200-2011

| Training | Subexplanation Counts Across Softmax Confidence Ratios | | | |
| | $\geq 80\%$ | $\geq 70\%$ | $\geq 60\%$ | $\geq 50\%$ |
|---|---|---|---|---|
| Standard | 11.6 | 26.4 | 43.0 | 67.6 |
| $L^-(0.7, 1)$ reg. | 59.9 | 96.3 | 127.0 | 164.0 |

Tab. 4 reports subexplanation counts of ResNet50 on CUB-200-2011 across varying softmax confidence ratios. As with ImageNet, training with $L^-(0.7, 1)$ regularization substantially increases the counts compared to standard training, indicating a shift toward compositional mid-order reasoning.

## A.5 ADDITIONAL PART-BASED EVALUATION OF INTERPRETABILITY ON CUB-200-2011

While the main paper evaluates interpretability in terms of prototype overlap with the annotated object bounding boxes, here we provide an additional part-based analysis leveraging the richer annotations available in CUB-200-2011. Specifically, since CUB includes keypoint annotations for 15 bird parts, we can assess alignment at a finer granularity that corresponds to semantically meaningful object components—especially important in fine-grained classification settings where recognizing characteristic object parts (e.g., beak, wing, tail) is essential.

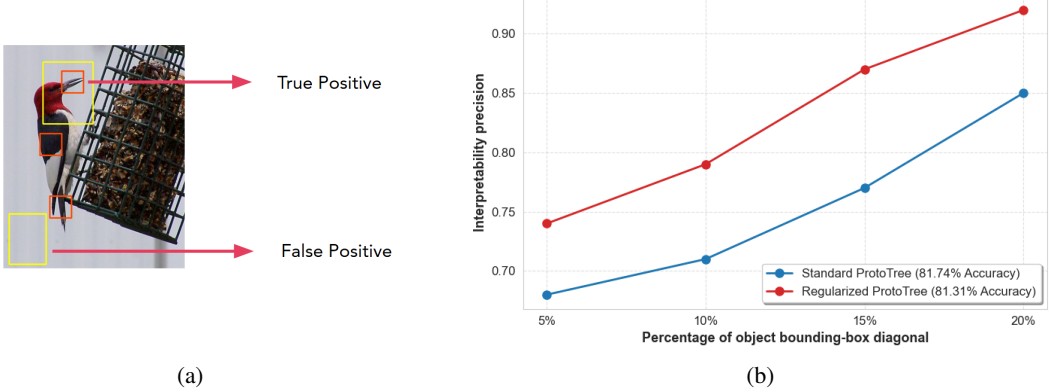

|       (a)       |       (b)       |

Figure 18: (a) Example of true and false positives for the part-based interpretability metric. Red boxes are ground truth examples - body part bounding boxes for CUB-200-2011. Yellow boxes are prediction examples (mapped prototypes). (b) Part-based interpretability precision comparison (IoU ¿ 0%) of mapped prototypes for standard and regularized ProtoTree.

For this analysis, we construct square part regions centered at each annotated keypoint. The side length of each square is scaled relative to the object size by setting its diagonal to 5%, 10%, 15%, and 20% of the object's bounding box diagonal. This scaling allows us to systematically evaluate how closely each prototype activation aligns with semantically meaningful bird parts across varying levels of spatial tolerance. As illustrated in Fig. 18a, yellow boxes indicate prototype activations, and

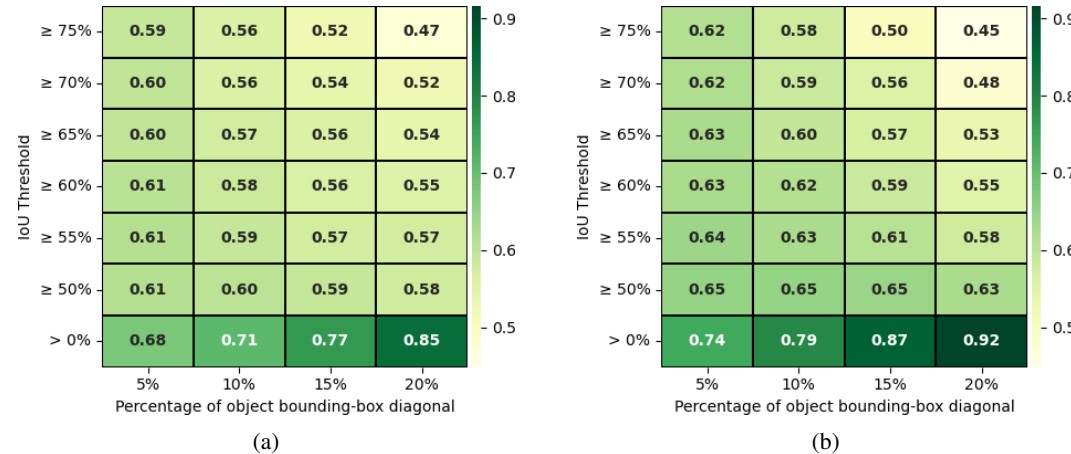

Figure 19: Part-based interpretability precision comparison of mapped prototypes for (a) standard and (b) regularized ProtoTree on CUB-200-2011

red boxes denote ground-truth part regions. The top activation overlaps with a part and is counted as a true positive, while the bottom one falls outside all part boxes and is treated as a false positive.

Figure 18b presents part-based interpretability precision computed under a simplified evaluation criterion that counts any overlap between a prototype and a ground-truth part region (i.e., IoU ¿ 0%) as a true positive. This summary captures the general trend across different part box sizes, showing that the regularized ProtoTree consistently outperforms the standard version. While both models improve as part boxes grow, the regularized ProtoTree maintains a clear lead across all scales.

To more rigorously evaluate interpretability across different spatial tolerances and matching stringency, Figure 19 reports precision heatmaps for both models over multiple part box sizes (x-axis) and IoU thresholds (y-axis). For each prototype, we identify its top-matching image patch from the training set and evaluate its overlap with the annotated part regions. A true positive is counted if the prototype's matched region overlaps with any annotated part region at the specified IoU threshold. The regularized ProtoTree consistently achieves higher precision across nearly all settings, particularly at lower IoU thresholds and larger part boxes. While precision gains diminish as thresholds become more stringent, the regularized ProtoTree outperforms the standard one in the majority of conditions. It is slightly lower only in a few settings, such as when the part box diagonal is 15% of object bounding-box diagonal and IoU $\geq$ 75%, or when the diagonal is 20% with IoU $\geq$ 65%. These results reinforce our hypothesis that directing the model's attention toward smaller, more structured contextual scopes improves its ability to associate prototypes with specific, part-level structures. This effect is particularly beneficial for fine-grained recognition, where interpretability hinges on accurately attending to subtle, localized differences across categories.

## A.6 QUALITATIVE PROTOTYPE-TO-REGION ALIGNMENT EXAMPLES

To further illustrate the structured interpretability improvements introduced by multi-order interaction modeling, we provide additional qualitative comparisons of ProtoTree explanations on test images. For each example, we compare the decision path of the standard ProtoTree with that of the regularized ProtoTree. Each path consists of a sequence of prototypes matched to regions in the test image. As shown, the standard ProtoTree frequently activates on vague or background regions, whereas the multi-order ProtoTree tends to align with semantically meaningful bird parts, suggesting improved part-awareness and more grounded explanations.

Fig. 20 presents qualitative comparisons of ProtoTree explanations for two test images from ImageNet-200 and CUB-200-2011, highlighting the differences in prototype-to-region alignment between the standard and regularized ProtoTree models. Green boxes indicate matches that align with annotated foreground or part regions, while red boxes mark vague or misaligned activations. In the first example (20a), the standard ProtoTree misclassifies an ox image as a "space heater", with its decision path dominated by prototypes that match grass textures and irrelevant background features.

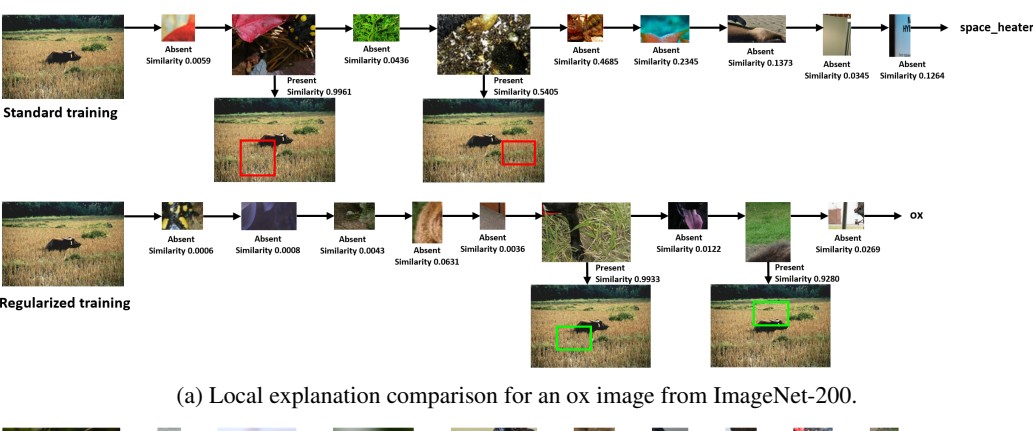

(a) Local explanation comparison for an ox image from ImageNet-200.

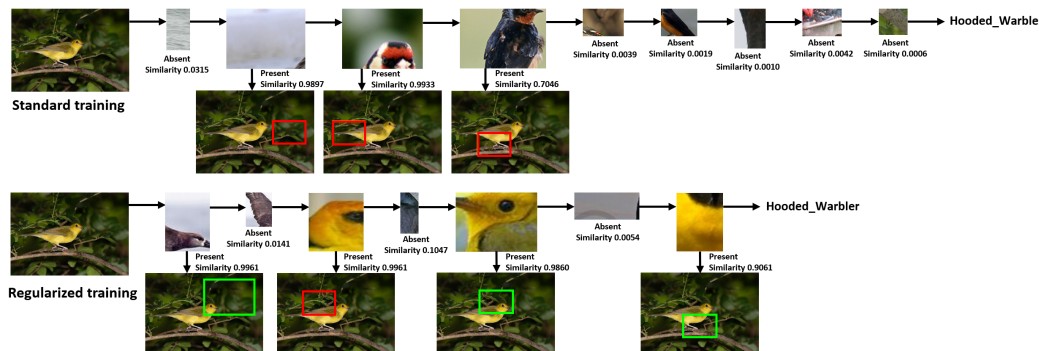

(b) Local explanation comparison for a Hooded Warbler bird image from CUB-200-2011.

Figure 20: Local explanations from standard and regularized ProtoTree models on (a) ImageNet-200 and (b) CUB-200-2011. Each explanation panel contrasts standard (top) and regularized (bottom) models. Green boxes indicate semantically meaningful, human-aligned prototype matches, while red boxes denote vague or misaligned activations. The regularized model yields more focused, part-relevant matches (e.g., fur texture and animal-on-grass for the ox, head and belly regions for the bird), while the standard model often attends to background or mismatched regions.

In contrast, the regularized ProtoTree aligns its prototypes with semantically meaningful parts of the object, including fur patterns and animal-on-grass regions, leading to the correct classification and a more coherent explanation. In the second example (20b), taken from CUB-200-2011, the standard ProtoTree again activates on background textures and mismatches a bird head prototype to the tail region. Meanwhile, the regularized ProtoTree matches two distinct head prototypes to the bird's head and a yellow body-pattern prototype to the yellow belly area, yielding more focused and biologically meaningful evidence.

Figure 21 provides local explanation examples for two test images from the CUB-200-2011 dataset, comparing the reasoning paths of the standard and regularized ProtoTree models. In the first example (21a,21b), the standard ProtoTree activates on background-like prototypes, one of which is matched to the bird's face in the test image, and another to an irrelevant background region. In contrast, the regularized ProtoTree routes its decision through prototypes that clearly represent class-relevant features such as the bird's belly and head, with matching patches in the test image showing strong semantic alignment. In the second example (21c,21d), the default ProtoTree initially routes the image through a series of vague or background-focused prototypes. One prototype activates on a background region in the test image, while another incorrectly matches the bird's body. In contrast, the regularized ProtoTree starts with a precise match between a prototype of a bird head and the corresponding region in the test image, followed by additional semantically consistent matches across the bird's body.

Overall, these examples demonstrate that suppressing extreme high-order interactions encourages ProtoTree to rely on more focused, part-aligned visual evidence. By steering decisions toward semantically meaningful regions and away from vague or background features, the regularized model

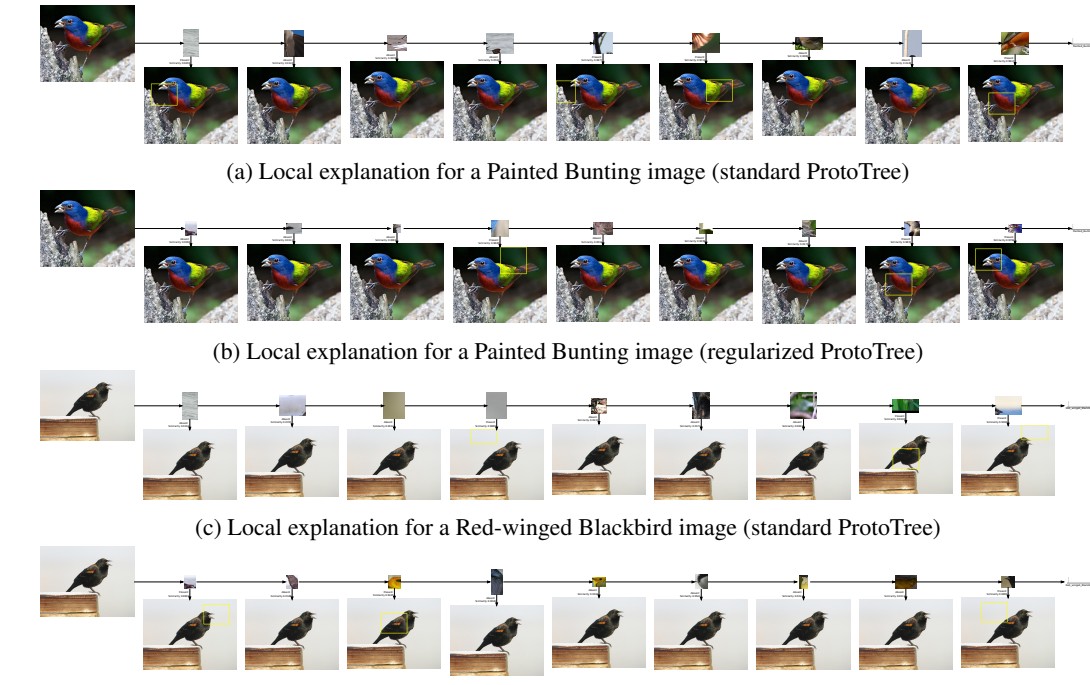

(a) Local explanation for a Painted Bunting image (standard ProtoTree)

(b) Local explanation for a Painted Bunting image (regularized ProtoTree)

(c) Local explanation for a Red-winged Blackbird image (standard ProtoTree)

(d) Local explanation for a Red-winged Blackbird image (regularized ProtoTree)

Figure 21: Local explanations from standard and regularized ProtoTree models on CUB-200-2011. Each explanation panel contrasts standard (top) and regularized (bottom) models.

produces explanations that are both more coherent and structurally grounded. This effect is particularly beneficial in fine-grained recognition tasks, where subtle part-level differences determine class distinctions, and it complements the quantitative improvements reported in the main paper.

## A.7    ADDITIONAL INTERACTION STRENGTH PLOTS FOR TRANSFORMERS

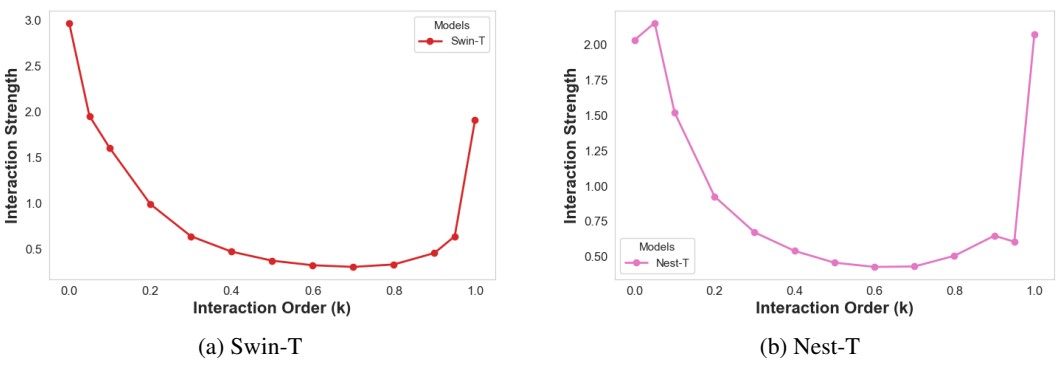

(a) Swin-T

(b) Nest-T

Figure 22: Multi-order interaction strength across for Swin-T and Nest-T on ImageNet.

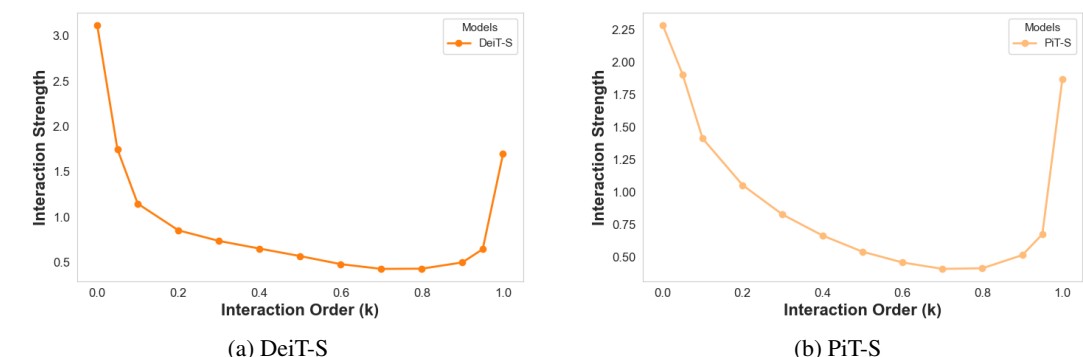

Figure 23: Multi-order interaction strength across for DeiT-S and PiT-S on ImageNet.

## A.8 THE USE OF LARGE LANGUAGE MODELS

Large language models (LLMs) are only used to aid or polish writing at the word or sentence level.

