# OpenReview forum: "Controlling Structured Explanations via Shapley Values"
_ICLR.cc/2026/Conference — Submitted to ICLR 2026_

### Official Review · Reviewer_EFeM · 2025-10-31

**Soundness:** 3
**Presentation:** 2
**Contribution:** 2
**Rating:** 4
**Confidence:** 3

**Summary:**

The paper proposes to first connect interaction Shapley values to "explanation scale", which is based on the number of explanations of different sizes, a specific prior explanation method.
After making this connection empirically, the paper analyzes a few architectures, particularly CNN and transformer based methods on their Shapley interaction values.
Like prior works, they note that CNNs focus on small details while transformers focus more on compositional structures.
The paper then suggests using previously proposed loss functions to train in tandem with the normal objectives to modify the model behavior.
The paper provides experiments showing they can modify the explanation behavior of models using these loss functions.

**Strengths:**

- The paper finds an interesting connection between interaction Shapley values and "explanation scale".
- The method does not require computing Shapley values or interaction strength directly.
- The paper demonstrates that it can modify the behavior of models using explanation regularizations.

**Weaknesses:**

- "Importantly, our goal is not to characterize or optimize the accuracy–explainability tradeoff, but to show how explanation structure can be systematically regulated while maintaining competitive accuracy, leaving a study of such tradeoffs to future work." This seems like an important qualification. But why do we want to regulate explanation structure if not for this type of trade-off? Merely showing that it's possible is not insightful unless it is *useful*.
  - "the value of controllability lies in adapting explanation structure to deployment needs, as prior work shows different structures can enhance robustness, modularity, or usability, and also improve user understanding and counterfactual reasoning". This suggests the potential benefits of controllability. However, none of these benefits are demonstrated. Only qualitative assessment of various explanations of model behavior. I would like to see results that clearly show this controllability enables one or more of these benefits for the task that is considered (when compared to relevant baselines for the task).

- The paper seems to follow Deng et al., 2022 a fair bit. And many of the sections have background and proposed methods interleaved. This makes it more difficult to isolate the contributions of this paper. Could you clearly mark all sections that are background? Ideally, all background would be in one section so that it is clear what is new and what is prior work. For example, though one of the main claims is about controlling explanation scale, the loss functions are from Deng et al. 2022. Thus, it is unclear what is novel here.
  - Overall, I would like a clearer separation. It seems that the main contribution is showing the connection and then using it to modify the model. But the actual loss functions and such are actually from a different paper.

- Computing intereaction strength reliably seems very difficult given that even Shapley values are hard to compute. Essentially, the variance of the estimators can be high given a small number of samples. Could you explain why the number of samples you chose was sufficient for reasonably accurate estimation?

- The correlation between subexplanation count and interaction strength doesn't seem particularly strong. Could you provide Pearson's, Spearman's and Kendall's Tau correlation values?

- "Explanation scale" is not a well-known term and needs to be at least intuitively explained  in the introduction. Also, the idea of "structured explanations" needs at least an example in the introduction as this is not a well-known term. I'm also a little concerned that this explanation type is fairly narrow and thus the impact of the paper is fundamentally based on this explanation method, which is not mainstream.

**Questions:**

- Is the connection between explanation scale and interaction strength surprising or unexpected? Could you give more discussion on this point, which is one of your main contributions? It seems that this is very natural. Are you merely saying that you give evidence for this or is there something more insightful that I'm missing?

- What are other ways to modify the behavior of a neural network? This seems to be the primary claim but it does not seem to be compared to alternatives in the prior work or other alternatives? It is hard to know if the proposed method is "good" at modifying the behavior if there is no comparisons.
  - For example, how does this compare to Shapley Explanation Networks [Wang et al., 2021]? This seems to be a related work specifically since it is used to control the training of the architecture but was not discussed in related works.

Wang, R., Wang, X., & Inouye, D. (2021, May). Shapley Explanation Networks. In International Conference on Learning Representations.

---

> ### Author Response · Authors · 2025-11-20
>
> We appreciate the feedback. We first address the questions (Q) and then the comments (C) regarding weaknesses.
>
> **_Q1: "Is the connection between explanation scale and interaction strength surprising or unexpected?"_**
>
> **A1:** Although the connection between explanation scale and interaction strength may feel intuitive once stated, it is not established or even hinted at in prior work (L90-96, L104-107). Structured explanations such as SAGs (Shitole et al., 2021) and Minimal Sufficient Subexplanations (Jiang et al., 2024) characterize explanation scale descriptively and identify architectural factors that influence compositionality, but they do not provide a general underlying principle explaining why different models produce explanations of different scales, nor do they connect explanation structure to the model's multi-order interaction strength (L89-91). Conversely, work on feature interactions—including Deng et al. (2022)—treats Shapley interactions purely as a diagnostic of model representations (L57-59), without linking them to explanation structure or showing that modifying them would affect how explanations behave.
>
> The non-obvious step in our work is to show that these two previously separate areas are governed by the same underlying quantity: multi-order Shapley interaction strength (L46-50). This relationship does not follow from existing definitions—SAG scale is defined through minimal sufficient subsets, while Shapley interactions arise from cooperative-game decompositions of model outputs. Their connection is not implied by the definitions themselves, it emerges from how the model actually combines features during inference. Our contribution is to (1) establish, within the Shapley interaction framework, the principled relationship between higher interaction strength and the number of valid subexplanations, (2) empirically validate this across diverse architectures, and (3) demonstrate that deliberately modifying interaction strength during training causally shifts explanation scale.
>
> Thus, we are not merely "observing a correlation", we are establishing a mechanistic link that was not recognized, used, or tested in prior interpretability or interaction-based training literature. We also show that this link enables a controllable training mechanism with practical consequences for structured explanations.
>
> **_Q2: "What are other ways to modify the behavior of a neural network?"_**
>
> **A2:** We appreciate the questions. As discussed in L91-94, prior work has explored explanation-based regularization to influence model predictions. For example, Ross et al. (2017), Ismail et al. (2021), and Plumb et al. (2020) apply regularization using human-defined or semantic cues to encourage models to focus on specific features, but these methods generally operate on isolated attributions and do not control structured explanation scales or reasoning patterns. Other global interaction measures such as Sobol indices (Fel et al., 2021; Iooss & Lemaître, 2015) decompose output variance but are designed for input-output attribution rather than shaping internal model reasoning (L101-104).
>
> Shapley Explanation Networks (Wang et al., 2021) integrate Shapley values as latent representations within a dedicated network architecture, enabling layer-wise explanation regularization. However, SHAPNETs require a new architecture to implement these Shapley modules, and thus are not architecture-independent. In contrast, as mentioned in L59-64, our method leverages multi-order Shapley interactions to causally manipulate the number and composition of valid subexplanations, consistently shifting structured explanation scales across CNNs, Transformers, and hybrid models without architectural redesign. This provides a principled, flexible mechanism to influence model reasoning that is not addressed by prior work. Empirically, we show that modifying interaction strength via our regularization reliably changes subexplanation counts, SAG scales, and ProtoTree alignment with object parts, demonstrating effective control over model behavior.

---

> ### Author Response · Authors · 2025-11-20
>
> **_C3: "But why do we want to regulate explanation structure if not for this type of trade-off?"_**
>
> **A3:** While we clarify that our work does not aim to explicitly characterize or optimize the accuracy–explainability tradeoff, the ability to regulate structured explanation scales is valuable in its own right. As discussed in L28-32 and L475-478, structured explanations provide insights into model reasoning, support user understanding, improve robustness, and enable alignment with human priors. By controlling interaction strength during training, our method allows targeted modification of model decision-making pathways, facilitating safe deployment, interpretability, and downstream reasoning tasks. Importantly, demonstrating that explanation structure can be systematically manipulated provides a concrete mechanism to influence model behavior—a contribution that stands on its own. While a formal study of accuracy–explainability tradeoffs is interesting, it requires a comprehensive analysis across a huge number of datasets, architectures, and regularization regimes, which is beyond the scope of this work. Our focus here is on demonstrating that structured explanation scales can be systematically regulated and that such regulation meaningfully influences model reasoning, leaving a full exploration of tradeoffs to future work.
>
> **_C4: "Only qualitative assessment of various explanations of model behavior"_**
>
> **A4:** The claim that our paper provides "only qualitative assessment" is factually incorrect. We provide multiple quantitative evaluations demonstrating that controllability meaningfully changes model reasoning, including (i) subexplanation counts and interaction-strength measurements (L319–323, Fig. 3, Tab. 1, Fig. 5), additional analysis for fine-grained classification (Appendix A.4, Tab 2.), (ii) saliency-removal robustness scores (L348-365, Tab. 2), (iii) prototype–foreground overlap precision for ProtoTree on CUB-200-2011 (Fig. 7), additional part-level analysis (Appendix A.5, Fig. 18-19) and (iv) additional quantitative interaction-strength analyses across 10+ architectures (Appendix A.3; Figs. 10–17). These directly show that adjusting interaction strength produces measurable, systematic shifts in explanation structure, not merely qualitative differences.
>
> The request to additionally show benefits in robustness, modularity, usability, or counterfactual reasoning through application-level evaluations is outside the scope of our contribution. Each of these constitutes a separate applied research direction requiring task-specific pipelines, baselines, metrics, and, in some cases, user studies. Our paper is a methodological and empirical work whose goal is to show that explanation structure can be systematically controlled during training (L59-64), and we already provide extensive quantitative evidence through multiple interpretability metrics that such control produces reliable, measurable changes in model reasoning. These forms of evaluation are standard for methodology papers establishing a new mechanism, and they directly validate the technical contribution. We do not claim that controllability automatically improves every possible downstream application; rather, we demonstrate that explanation structure is now a trainable dimension of model behavior. This capability is valuable on its own and provides the foundation for future task-specific investigations.

---

> ### Author Response · Authors · 2025-11-20
>
> **_C5: "many of the sections have background and proposed methods interleave"_**
>
> **A5:** We clarify that the placement of background material is intentional. The paper relies on several distinct interpretability tools—SAGs, subexplanation counts, multi-order interactions, and ProtoTrees—which differ substantially in definitions and notation. Consolidating all background into a single section would force readers to continuously cross-reference material and disrupt the flow of the experimental sections. Instead, we provide the general background in Sec. 2 and 3 and introduce additional definitions precisely where they are required, a common structure for methodology papers involving multiple frameworks.
>
> On novelty, the paper clearly states (L57–59) that Deng et al. (2022) use the interaction-strength loss only to analyze a CNN representational bottleneck. They do not study transformers or hybrid architectures, do not relate interaction strength to explanation scale, and do not examine controllability of structured explanations. In contrast, our work extends the analysis across a broad range of architectures—including hierarchical transformers, hybrids, and distilled models—with detailed per-model interaction-strength visualizations across orders and confidence thresholds (L233–235, Appendix A.3), and shows that the representation bottleneck is architecture-dependent (L226–233) rather than universal as claimed by Deng et al. (2022). Moreover, our contributions include demonstrating a consistent, architecture-wide correlation between subexplanation counts and Shapley-based interaction strength, establishing interaction structure as a principled lens on model reasoning (L216–223), and addressing limitations of prior explanation metrics and regularizers by connecting explanation scales to multi-order Shapley interactions, enabling principled explanatory regularization (L89–96). We further evaluate this controllability across multiple interpretability frameworks (Sections 5.2–5.3, Fig. 7, Appendices A.4–A.5). Although the loss functions are adopted, the scientific questions, analyses, and resulting contributions are fundamentally different.
>
> **_C6: "Computing intereaction strength reliably seems very difficult"_**
>
> **A6:** We follow standard, empirically validated sampling from prior work (L183–187). Ablations in Appendix A.1, Fig. 8 show that with the choice of number of samples, the instability of $J^{(k)}$ drops below 0.15, indicating a stable and reliable estimate. Larger sampling gives minimal gains but incurs prohibitive storage and compute, while smaller sampling increases variance. Note that this estimation is only required for analysis. As mentioned in L269-272, the interaction-based training losses do not compute Shapley values at training time and add only modest overhead.
>
> **_C7: "Could you provide Pearson's, Spearman's and Kendall's Tau correlation values"_**
>
> **A7:** To address the request, we computed the correlation values between interaction strength $J^{(k)}$ and the average subexplanation counts across all 13 models in Fig. 1 of the paper. The metrics show a moderate linear correlation (Pearson = 0.47) and strong monotonic trends (Spearman = 0.66, Kendall = 0.51), confirming that models with higher interaction strength consistently exhibit higher subexplanation counts. These results quantitatively support the trend already shown in Fig. 1 and reinforce our claim that explanation scale is tightly coupled to the strength of feature interactions a model encodes. The values have been added to the revised manuscript.
>
> **_C8: "Explanation scale is not a well-known term and needs to be at least intuitively explained in the introduction"_**
>
> **A8:** We explicitly introduce and describe structured explanations in the introduction (L26–32) as explanations capturing both local saliency and global reasoning pathways, with textual examples in the form of SAGs (no figure due to space constraints). "Explanation scale" is also defined (L48–51) when discussing different granularity levels of explanations and their connection to subexplanation counts and feature-interaction strength. To improve clarity for all readers, we have slightly revised the wording to make this connection more explicit. While our work focuses on structured explanations—a well-defined class, as highlighted in the paper title, "Controlling Structured Explanations via Shapley Values"—our methodology generalizes across multiple frameworks (SAGs, ProtoTrees, subexplanation counts, saliency distributions), demonstrating broad applicability within this class. The concern that the approach is "not mainstream" ignores that structured explanations themselves are a principled, widely studied class for capturing model reasoning, not a single isolated method.

---

### Official Review · Reviewer_WxPK · 2025-10-31

**Soundness:** 2
**Presentation:** 2
**Contribution:** 2
**Rating:** 4
**Confidence:** 3

**Summary:**

The paper introduces Shapley-value-based method for providing structured explanation for neural network. The key advantage is to incorporate interaction-based regularization into training. It considers multi-order feature interaction strength for explanation structure, enabling models to be trained for desired reasoning behaviors

**Strengths:**

- It proposes a novel approach for incorporating interaction interactions during training
- The evaluation was performed across multiple model types and datasets.

**Weaknesses:**

- Compromised accuracy would be barrier for practical adoption of this method.
- Trade-off introduced by the method, such as stability vs storage, can be hard to tune in practice without rigorous analysis each time.

**Questions:**

- Can this regularization method enhance model robustness or out-of-distribution generalization?

---

> ### Author Response · Authors · 2025-11-20
>
> We appreciate the feedback. We first address the questions (Q) and then the comments (C) regarding weaknesses.
>
> **_Q1: "Can this regularization method enhance model robustness or out-of-distribution generalization?"_**
>
> **A1:** We clarify that our method is not designed to directly improve robustness to out-of-distribution generalization. Rather, it aims to control the structured explanation via multi-order interactions—encouraging more compositional or disjunctive reasoning through Shapley interaction regularization. While our analyses include semantic alignment evaluations (e.g., prototype-to-region behavior in ProtoTree), these do not imply that the regularizer directly enforces semantic meaning. Instead, they show that shaping the interaction structure can affect interpretability and alignment of model explanations—even without semantic supervision. We do not evaluate under distribution shift or OOD settings, which are beyond the scope of this work. However, promoting meaningful reasoning may benefit robustness, and we consider this a promising direction for future research.
>
> **_C2: "Compromised accuracy would be barrier for practical adoption of this method."_**
>
> **A2:** As mentioned in L63–65, we clarify that the paper does not aim to characterize or optimize the accuracy–explainability tradeoff. Our contribution is to demonstrate that structured explanation scales can be systematically controlled through Shapley-based interaction regularization, which is valuable independently of accuracy tuning because it enables targeted modification of a model’s reasoning behavior, alignment with human priors, and safer deployment. Showing that explanation structure can be explicitly manipulated provides a concrete mechanism for influencing model behavior—a contribution that stands on its own. A full study of accuracy–explainability tradeoffs would require extensive evaluation across many datasets, architectures, and regularization regimes, which is beyond the scope of this work. We therefore focus on establishing controllability of explanation structure, leaving a comprehensive analysis of accuracy tradeoffs to future work.
>
> **_C3: "Trade-off introduced by the method, such as stability vs storage, can be hard to tune in practice without rigorous analysis each time."_**
>
> **A3:** The mentioned stability–storage trade-off concerns only the offline estimation of Shapley interaction strengths $J^{(k)}$ for analysis (Appendix A.1) and is not part of training. As described in Section 4.2, L269-272, the Shapley-based interaction losses $L^+$ and $L^-$ are computed using masked input variants and do not require computing Shapley values, adding only a modest linear overhead per epoch. The sampling parameters used to evaluate $J^{(k)}$ were chosen to ensure stable measurements and have no impact on training or practical deployment.

---

> > ### Comment · Reviewer_WxPK · 2025-11-27
> >
> > Thank you for the detailed response, particularly the clarification regarding the computational overhead and the scope of the work. However, I remain concerned about the practical adoption barrier due to compromised accuracy. While the authors state the paper is not optimizing the accuracy–explainability trade-off, a large, or even an unquantified, drop in performance will prevent this technique from being integrated into real-world applications. I think better characterizing the performance drop comprehensively would be very helpful in order to ensure the practical impact.

---

### Official Review · Reviewer_MJVm · 2025-10-31

**Soundness:** 3
**Presentation:** 3
**Contribution:** 3
**Rating:** 6
**Confidence:** 4

**Summary:**

The paper proposes a method to control the structured explanations obtaining during network training by introducing a framework regularization based on Shapley values. The authors analyze Transformer and CNN architectures using this framework.

**Strengths:**

- The use of  Shapley values to encode varying levels of feature dependence providing greater control over how the network leverages the features.
- The experiments with ProtoTree that demonstrate  the proposed regularization substantially improve interpretability.

**Weaknesses:**

- **Motivation for control:** I like the idea of having control over interactions, but I’m not sure why this is necessary. What is the underlying motivation for introducing this control? As you mentioned, if there isn’t a clear notion of an “optimal” level, what problem does this control solve?
- **Discussion on oversimplification:** You mention that explanations can be oversimplified, and I agree that neural networks are inherently complex. However, increasing the complexity of explanations does not necessarily improve interpretability. In fact, one could say that providing more distinct important regions can make interpretability more difficult.
- **Clarification of SAG visualizations:** I’m having trouble interpreting the visualizations in Figure 4 — the highlighted regions (red segments) look almost identical across multiple repetitions. Since SAG explanations appear central to your analysis, it would be helpful to provide more details on how these visualizations should be interpreted.

**Questions:**

- **Experiment in Table 2:** I found the experiment in Table 2 interesting. Which dataset was used? Do you observe similar trends across other datasets? Also, what is the rationale for focusing on reducing the importance of localized regions? Why aim for sparse explanations? While you propose an additional level of control, could this potentially make interpretation more difficult rather than easier?
- **Patch features:** Since patches are used as features, does the patch size affect the results? Would it be feasible to use features corresponding to human-interpretable concepts instead?
- **Effect of regularization on high-order interactions:** Why does regularizing to reduce high-order interactions help ProtoTree focus on the foreground? What would happen if the regularization were applied in the opposite direction, for instance to reduce the L+ term?
- I also include some questions above.

---

> ### Author Response · Authors · 2025-11-20
>
> We appreciate the feedback. We first address the questions (Q) and then the comments (C) regarding weaknesses.
>
> **_Q1: "Experiment in Table 2 ... why aim for sparse explanations?"_**
>
> **A1:** As mentioned in L351-354, Tab. 2 follows the experimental setup of Ismail et al. (2021) exactly. We use their CNN architecture, and training protocol on the MNIST dataset, adding only the $L^-$ loss. In their work, the authors restrict this experiment to MNIST because the deletion-based metric requires a known uninformative background, which is not available for natural-image datasets. Therefore, we do not apply this specific deletion metric elsewhere. However, we observe consistent trends across our other interpretability analyses. For example, in L291-338, interaction regularization for ResNet50 on ImageNet increases subexplanation counts and produces longer, less localized attribution paths in SAG. This reduced reliance on concentrated regions aligns with the behavior observed in Tab. 2.
>
> Importantly, our goal is not to assert that sparse or less‐localized explanations are inherently superior (L407–409), but to provide a mechanism for controlling explanatory structure. Reducing overreliance on small localized regions is one way to control explanatory structure. This mirrors the motivation in Ismail et al. (2021), where models depending on a few highly salient pixels are brittle under occlusion or corruption, with explanations dominated by isolated artifacts. Encouraging a more distributed interaction pattern can mitigate this brittleness, as reflected in Tab. 2 where models trained with $L^-$ degrade more gradually under deletion. At the same time, excessively diffuse explanations can indeed be harder to interpret, which is precisely why our method is designed as a controllable knob rather than a normative prescription: by adjusting $(k_1, k_2)$, users can induce either more concentrated or more distributed explanations. Our contribution is to expose this control, not to claim that one level of sparsity or localization is universally preferable. This aligns with our broader framing throughout the paper (e.g., L369: "if such behavior is desired"), where interaction patterns are treated descriptively rather than normatively.
>
> **_Q2: "Since patches are used as features, does the patch size affect the results?"_**
>
> **A2:** We break images into patches when computing interactions and applying regularization because computing interactions at the pixel level would be computationally infeasible. The patch size defines the granularity of these interactions: smaller patches allow finer control, while larger patches reduce computational cost. During training, the model receives masked variants of the patch-level input to compute Eq. 4, and the corresponding losses in Eq. 5, so the regularization operates on patch-based features rather than individual pixels. In principle, one could use features corresponding to human-interpretable concepts (e.g., object parts or semantic attributes) instead of patches, but this would require a predefined concept set or annotations, which is outside the scope of our current study.
>
> **_Q3: "Effect of regularization on high-order interactions"_**
>
> **A3:** In ProtoTree, each prototype makes soft routing decisions by comparing itself to all patches in an image and selecting those with the highest similarity (L414–416). We believe that suppressing high-order interactions effectively reduces the number of patches that contribute to each decision, which likely diminishes background clutter while preserving informative foreground patches (L420-423). Since most datasets contain more background than foreground, randomly masking patches is more likely to remove background, allowing prototypes to focus relatively more on informative foreground regions. Occasionally, foreground patches are also masked, but this mechanism plausibly explains the observed increase in foreground alignment precision (Fig. 7), and frames $L^-$ regularization as guiding, rather than enforcing, more interpretable prototype behavior. Applying $L^+$ to encourage high-order interactions would likely diffuse prototype activations across many patches, including background, potentially reducing focus on semantically meaningful regions.

---

> ### Author Response · Authors · 2025-11-20
>
> **_C4: "What is the underlying motivation for introducing this control?"_**
>
> **A4:** Thank you for raising this point. As discussed in L27–32, L52–64, and L165–171, prior work, e.g., Jiang et al. (2024), Shitole et al. (2021), has shown that explanation structure plays a significant role in robustness, generalization, and interpretability. Our contribution is to provide a principled mechanism for modulating this structure during training, without modifying the architecture or requiring additional supervision. This capability is valuable because, as prior works demonstrate, explanation structure directly affects model behavior and user understanding. For example, Jiang et al. (2024) shows that promoting higher-order (compositional) interactions improves robustness to occlusion and noise, while encouraging lower-order (disjunctive) interactions supports modularity and facilitates debugging. Likewise, Shitole et al. (2021) introduce Structured Attention Graphs and find that users answer counterfactual queries more accurately and confidently when given structured rather than unstructured explanations. Since these needs vary across deployment settings, there is no single "optimal" interaction level.
>
> Thus, as also discussed in L474-478, the key benefit of our approach is enabling flexible, targeted adjustment of explanatory behavior to align with deployment-specific needs, such as safety, redundancy, or sparsity. Rather than committing to a fixed explanatory profile, our method offers adaptability across tasks and operating conditions. As noted in L481–485, we view this as a promising direction for interpretability research: moving from passive interpretation of what a model has learned to proactive control over how a model reasons.
>
> **_C5: "Discussion on oversimplification"_**
>
> **A5:** We agree that neither oversimplified nor overly complex explanations are inherently more interpretable. In the paper, we motivate structured explanations, such as Structured Attention Graphs (Shitole et al., 2021), as a richer alternative to conventional saliency maps (L24–32). SAGs systematically capture both local saliency and global reasoning pathways, and their utility is supported by a large-scale user study (100 participants) in their paper, demonstrating that participants answer counterfactual questions more accurately and with better-calibrated confidence than when using standard saliency maps. Our contribution is complementary: we provide a controllable mechanism that modulates the structure of explanations during training (L51–53, 61–63), allowing practitioners to adjust whether reasoning appears more concentrated (lower-order) or distributed (higher-order) depending on the interpretability needs of the application. We do not claim that a more complex structure is always better; rather, our framework treats explanation structure as a design choice (L369) informed by the deployment context, consistent with the broader motivation for structured explanations.
>
> **_C6: "Clarification of SAG visualizations"_**
>
> **A6:** Thank you for pointing this out. We clarify that in SAG visualizations the red regions indicate the patches *removed* at each node to form a valid subexplanation, not the patches attended to by the model. This is described in Sec. 4.1 (L160–163), where we explain that SAG is constructed by iteratively removing subsets of the Minimal Sufficient Explanation $S$ while maintaining a minimum confidence threshold. Because the red region is removed from the image, the area underneath it is necessarily identical across nodes. Each node visualizes the model's softmax confidence score—relative to the score on the full image—when that specific region is removed. The purpose of Fig. 4 is to illustrate that under regularization, the model maintains high confidence even after more regions are removed, resulting in a larger SAG with more valid subexplanations. We have revised the caption to clarify this.

---

### Official Review · Reviewer_wwJS · 2025-10-31

**Soundness:** 3
**Presentation:** 2
**Contribution:** 2
**Rating:** 6
**Confidence:** 2

**Summary:**

The paper introduces a Shapley value–based interaction regularization that integrates interpretability objectives into model training. The method enables controllable shifts in reasoning style and explanation scale across architectures such as CNNs and Transformers.

**Strengths:**

- The paper provides interesting empirical evidence that CNNs exhibit more disjunctive (localized) reasoning, while Transformers demonstrate more compositional (holistic) reasoning. This comparison helps clarify architectural differences in how models integrate features and reason over spatial contexts.
- By incorporating Shapley-based interaction regularizers into the training objective, the paper moves beyond post hoc analysis and offers a method to influence interpretability during learning.
- The experiments show that adjusting the regularization strength can shift models between holistic and localized reasoning, illustrating that the method offers controllable interpretability behavior rather than fixed outcomes.

**Weaknesses:**

- The paper claims to "establish a theoretical connection" (line 43) between subexplanation counts and Shapley-based interaction strength, but the evidence presented appears to be empirical and very weak.
- The novelty appears limited. The proposed multi-order interaction regularizer completely follows Deng et al. (2022). The main contribution seems to lie in the empirical observation of differences in interaction structures between CNNs and Transformers, and this work looks like a simple combination of Jiang et al. (2024) and Deng et al. (2022).
- The paper could benefit from improved clarity and explanation of figures. For readers less familiar with the specific background, the lack of a comprehensive description makes it difficult to understand how the figures are drawn. This makes it challenging to verify the authors’ claims.
- The paper presents an internal inconsistency regarding the role of interaction-based regularization. In lines 407–409, the authors state that the proposed method merely provides controllability over the model’s reasoning behavior "without implying that any particular style is inherently superior or more interpretable." However, in lines 455–457, the authors claim that their regularization leads to more semantically meaningful representations and improves the accuracy of explanations.

**Questions:**

- Could the authors clarify the exact meaning of the axes in Figure 2? In addition, I noticed that the ranking of models appears to vary across different orders. Does this variability suggest any limitations in the stability or reliability of the proposed interaction-based evaluation？
- In Eq. (4), the computation of $U_c(k_1, k_2)$ involves an expectation over masked subsets. However, the paper states that the training loss "only involves an additional forward–backward pass on two masked variants of the input." Could you clarify how this expectation is handled during optimization?
- The paper uses different ranges of $k$ across figures (e.g., 0.05–0.5 in Figure 1, 0.1–0.4 in Figure 2, and >0.5 in Figure 3). Could the authors clarify how these $k$ values are chosen in practice?
- I am somewhat confused by the discrepancy: Figure 4 suggests that CNNs mainly focus on localized regions, which would imply stronger interactions at lower $k$. However, in line 300 and Figure 3, the interaction strength for CNNs appears concentrated at higher $k$. Could the authors reconcile these observations or explain whether the definition or scaling of k differs across figures?

---

> ### Author Response · Authors · 2025-11-20
>
> We appreciate the feedback. We first address the questions (Q) and then the comments (C) regarding weaknesses.
>
> **_Q1: "Could the authors clarify the exact meaning of the axes in Figure 2?"_**
>
> **A1:** As defined in L127–142, and described in the caption of Fig. 2 and L224-227, the x-axis in Fig. 2 corresponds to the interaction order $k$, which specifies the proportion of contextual features considered when computing pairwise Shapley interactions $\phi^{(k)}(i,j)$. The y-axis denotes the corresponding multi-order interaction strength $J(k)$, i.e., the aggregated magnitude of these pairwise interactions over all features and images. We have revised the figure by explicitly labeling the x-axis as "Interaction Order (k)".
>
> The variability of interaction strength $J(k)$ across orders  $k$ is central to our analysis—it reflects genuine architectural differences rather than instability. As discussed in L224–234, CNNs exhibit stronger interactions for small $k$ and weaker interactions for mid-order $k$ values, while Transformers capture pronounced interactions over a range of mid-order values  (0.1 < $k$ < 0.4) due to their broader contextual modeling via self-attention. Importantly, this trend is stable across runs and further corroborated by per-model heatmaps in Appendix A.3.
>
> **_Q2: "In Eq. (4), the computation of $U_c(k_1, k_2)$ involves an expectation over masked subsets"_**
>
> **A2:** The expectation is approximated via Monte-Carlo sampling. For each training iteration, we draw one pair of random subsets $(S_1, S_2)$ for each image in the batch by generating two independent dropout masks (corresponding to $k_1$ and $k_2$). These produce two masked inputs, on which we compute $U_c(k_1, k_2) = \text{softmax} \Big( z_c(S_2) - \frac{k_2}{k_1} z_c(S_1) \Big)$ using the additional passes, and backpropagate through this estimate. This follows the approximation scheme of Deng et al. (2022).
>
> **_Q3: "The paper uses different ranges of $k$ across figures"_**
>
> **A3:** The ranges of $k$ in Fig. 2 and 3 are the same. Both figures show the full range of $k \in [0,1]$. The caption in Fig. 2 simply highlights the sub-range $[0.1, 0.4]$ where the model differences are most pronounced. In Fig. 1, for clarity, we show an interval of $k \in [0.05, 0.5]$. As mentioned in L216-219, we focus on the interval $k \in [0.05, 0.5]$ in Fig. 1 to highlight the largest architecture-dependent differences. The underlying computation for the plot in Fig. 1 still spans the full range $k \in [0,1]$.
>
> **_Q4: "I am somewhat confused by the discrepancy"_**
>
> **A4:** We're happy to clarify. As described in L224-234, CNNs exhibit the "representation bottleneck" (Deng et al., 2022), whereby low-order interactions are strongest, mid-order interactions are weak, and high-order interactions have a secondary peak. The underlying reason, derived in Sec. 3.2 of Deng et al., is that the training strength of $k$-order interactions is proportional to $\frac{|N|-k-1}{|N|(|N|-1)} \Big/ \sqrt{\binom{|N|-2}{k}}$, which is higher for small or large $k$ and much lower for intermediate $k$. Consequently, CNNs encode strong low-order interactions in localized regions while also exhibiting a modest high-order peak when most patches are present (Fig. 3(a) for ResNet50, similar trends for VGG19 are shown in Appendix A.3, Fig. 10). The definition and scaling of $k$ is consistent across all figures.

---

> ### Author Response · Authors · 2025-11-20
>
> **_C5: "The paper claims to establish a theoretical connection"_**
>
> **A5:** Thank you for pointing this out. To clarify, the text in L43 was intended to highlight that subexplanation counts can be interpreted through the principled framework of Shapley values, which formally quantify feature interactions. Our empirical results show a consistent correlation between subexplanation counts and Shapley-based interaction strength across models and spatial contexts, supporting this principled interpretation. To make this evidence more explicit, we have now added the computed correlation coefficients to the paper (Pearson = 0.47, Spearman = 0.66, Kendall = 0.51). We have also revised the sentence for clarity.
>
> **_C6: "The novelty appears limited."_**
>
> **A6:** We respectfully disagree that our work lacks novelty. As described in L57-59, prior work applied the regularization loss mainly to CNNs and only to analyze the CNN's representational bottleneck, without studying how modifying interaction strength affects explanation scales or reasoning patterns. In contrast, our work:
>
> - Extends the analysis beyond CNNs to a wide spectrum of architectures, including hierarchical transformers, hybrids, and distilled models, with comprehensive per-model visualizations of interaction strength patterns across orders and confidence thresholds (L233-235, Appendix A.3).
> - In L226-233, reveals that the representation bottleneck described by Deng et al. (2022) is, in fact, architecture-dependent rather than universal, as they claim.
> - Demonstrates a consistent, architecture-wide correlation between subexplanation counts and Shapley-based interaction strength (L216–223), establishing interaction structure as a principled lens on model reasoning.
> - Addresses the limitations of prior explanation metrics and regularizers (L89–96): sub-explanation counts do not clearly reveal the underlying reasoning principles or offer guidance on how to modify them, and prior regularizers operate only on isolated attributions. Our formulation connects explanation scales to multi-order Shapley interactions, enabling principled explanatory regularization.
> - Evaluates the effects of interaction-based regularization across multiple interpretability frameworks—SAGs, sub-explanation counts, saliency distributions, and ProtoTrees, including both quantitative and qualitative analyses (Sections 5.2 and 5.3).
> - Provides additional evaluations on fine-grained CUB-200-2011 dataset, covering both subexplanation counts and part-based ProtoTree interpretability (Fig. 7, Appendices A.4–A.5).
>
> Together, these contributions go beyond simply observing interaction differences or combining prior methods. They establish a principled framework for actively shaping structured explanations across diverse models, which has not been previously addressed.
>
> **_C7: "The paper could benefit from improved clarity and explanation of figures"_**
>
> **A7:** We appreciate the suggestion. The contents of Figs. 1–2 are described in L216–235, Fig. 3 in L297–300, Fig. 4 in L316–320, and Figs. 5–6 in L369–374, with Fig. 7 explained in detail in L425–462. The figure captions are already fairly detailed, and we were limited in extending them further due to space constraints. We believe the main text provides sufficient explanation for readers to interpret the figures and verify the claims.
>
> **_C8: "The paper presents an internal inconsistency"_**
>
> **A8:** In L407–409, our statement refers to the general mechanism: interaction-based regularization provides control over a model's reasoning behavior (e.g., more distributed vs. more modular), and we explicitly avoid claiming that any reasoning style is universally superior or inherently more interpretable. This aligns with our broader framing throughout the paper (e.g., L369: "if such behavior is desired"), where interaction patterns are treated descriptively rather than normatively.
>
> In contrast, L455–457 discusses a specific, task-grounded evaluation within the ProtoTree framework. Here, "more semantically meaningful representations" refers to annotation-based alignment—i.e., prototypes activating more on ground-truth foreground regions or semantic parts (e.g., beak, wing, tail) on ImageNet-200 and CUB-200-2011. This does not claim a universal notion of interpretability nor superiority of a reasoning style, rather, it reflects improved alignment under a well-defined interpretability metric inherent to ProtoTree.
>
> Thus, these sections address different questions—general controllability versus metric-grounded evaluation—and are fully consistent when viewed within their respective scopes.

---

> > ### Comment · Reviewer_wwJS · 2025-11-25
> >
> > Thank you for the detailed response. It helped clarify several points I did not fully understand earlier. I also took this opportunity to read the work of Deng et al. (2022) (which I apologize for not reading before the initial review — my first-round comments were solely based on the content presented in your paper).
> >
> > After reading it, I now have several concerns:
> >
> > **1. Concerns about the claim in Lines 226–233**
> >
> > From my reading, I feel that the “universal” conclusion in Deng et al. (2022) still holds. As you explained in A4 (which corresponds to Theorem 1 in Deng et al. (2022)), all models tend to exhibit strong low-order and high-order interaction strengths, and weak middle-order strengths, as also shown in your Figure 2.
> >
> > You mentioned that Transformer-based architectures show stronger middle-order interactions, but your chosen range for k is 0.1–0.4. In contrast, Deng et al. (2022) define middle-order interactions as covering a much wider range (0.1–0.9). When I examine your curves in the second half of that interval (0.5–0.9), I actually observe that CNN architectures appear to have slightly stronger interaction strengths than Transformer architectures. Could you clarify why this happens?
> >
> > Additionally, could you explain why at k=0 the interaction strengths of Swin-T and ConvNeXt-T are noticeably lower than ResNet-50 and DeiT-S-dis? A similar question applies to k=1, where Swin-T shows the highest interaction strength while ConvNeXt-T ranks third. These differences are interesting but not intuitively explained in the paper.
> >
> > **2. Regarding Q2 (one-shot estimation)**
> >
> > Since your loss formulation follows Deng et al. (2022) closely, I would like to confirm whether Deng et al. (2022) also use a single-sample (one-shot) estimation. Do you believe that one-shot estimation is still reliable when S becomes large? From my understanding, in Shapley-value-based explanation frameworks, approximating expectations using only one Monte Carlo sample can be risky and may lead to extremely noisy estimates.
> >
> > **3. Choice of k range in Figure 1**
> >
> > Given that k is defined on the interval [0, 1], I suggest plotting the entire range in all relevant figures, including Figure 1. The current choice of k=0.05–0.5 feels somewhat weird. It may be clearer and more consistent to follow the ranges in Deng et al.(2022), such as 0–0.1 (low-order), 0.1–0.9 (middle-order), and 0.9–1.0 (high-order).
> >
> > Finally, I have updated my confidence score (from 2 to 3) and slightly lowered my overall score (from 6 to 4). I look forward to your response, which may further lower my score to 2 — or potentially bring it back to 6 or rasing it to 8.

---

> > > ### Author Response · Authors · 2025-11-27
> > >
> > > We appreciate the additional comments. We address them in A9–A12.
> > >
> > > **_C9: "Concerns about the claim in Lines 226–233"_**
> > >
> > > **A9:** We could not find any evidence in Deng et al. (2022) supporting the reviewer’s interpretation that the middle-order range is strictly $k\in[0.1,0.9]$. Instead, the paper describes this region qualitatively and uses different ranges in different experiments:
> > >
> > > - **Page 5, "Representation bottleneck."** They state that interaction strength is usually high when the order $(m < 0.1n)$ or $(m > 0.9n)$, and low "when the order $m$ approximates $0.5n$" (their wording). This is descriptive—not a formal definition of middle-order as the entire interval $[0.1,0.9]$.
> > >
> > > - **Page 8, Sec. 3.4.** To demonstrate that their loss functions can modulate interaction orders, they explicitly train a "middle-order" model by *boosting* interactions in $k\in[0.3,0.7]$. This setup contradicts the reviewer’s interpretation of middle-order as $[0.1,0.9]$. Likewise, their "low-order" model *penalizes* $[0.7,1.0]$ (not $[0.1,1.0]$), and their "high-order" model *penalizes* $[0.0,0.5]$ (not $[0.0,0.9]$). These choices indicate that Deng et al. do not treat these boundaries as fixed categories but adjust them based on the specific analysis.
> > >
> > > Given this, our focus on $k\in[0.1,0.4]$ follows the same spirit: we highlight the region where architectural differences are most pronounced (as noted in the Fig. 2 caption). Outside this region—especially for very high orders $k>0.5$—the bottleneck effect described by Deng et al. emerges. Crucially, its onset and shape are not universal: Transformers sustain stronger interactions over a broader range, whereas the small increase in ResNet-50 at high order is consistent with their theoretical curve and remains far weaker than the clear separation in the mid-order region we emphasize. To provide further evidence, we added additional Transformer plots in Appendix A.7, which make this distinction explicit.
> > >
> > > **_C10: "Could you explain why at k=0 the interaction strengths"_**
> > >
> > > **A10:** At $k=0$, only two patches remain active, so the model processes an extremely sparse input. A plausible explanation is that CNNs such as ResNet-50 can still extract non-trivial features because local convolutional filters operate effectively even on isolated patches. In contrast, Transformer attention becomes poorly conditioned when only a few informative tokens remain, yielding weaker interaction estimates. At $k=1$, where all patches are present, the trend reverses: Transformers naturally exploit global context via self-attention and therefore show stronger interaction strengths than CNNs. These interpretations align with known architectural biases and contextualize the edge cases at $k=0$ and $k=1$, consistent with prior observations (L166–171) that CNNs behave more disjunctively while Transformers exhibit more compositional behavior in structured explanations.
> > >
> > > **_C11: "Regarding Q2 (one-shot estimation)"_**
> > >
> > > **A11:** To clarify, this is not a one-shot estimation. As noted in A2, a new pair of dropout masks is sampled for every image in every iteration of every epoch. Over the course of training, each example therefore contributes hundreds of distinct Monte-Carlo samples. The approximation is therefore not based on a single sample, but accumulates over repeated stochastic sampling throughout training, which substantially reduces variance. This repeated stochastic sampling is exactly the approximation scheme used in Deng et al. (2022), whose training code is publicly available at: *https://github.com/Nebularaid2000/bottleneck* (see `trainer_dropout_deltav_baseline.py` and `trainer_dropout_entropy_deltav_baseline.py`).
> > >
> > > **_C12: "Choice of k range in Figure 1"_**
> > >
> > > **A12:** The reviewer’s request is not technically meaningful. In Fig. 1, the x-axis is the mean interaction strength averaged over $k \in [0.05, 0.5]$. The reviewer asks us instead to aggregate over the full range $[0,1]$ and compare architectures using that value. However, as defined in Eq. (3), $J^{(k)}$ is normalized by the *mean over all orders*, which forces the global average across $k \in [0,1]$ to equal $1$ for every model. Consequently, averaging over the full interval collapses all models to the same value and cannot distinguish architectures. A zoomed view is therefore required to expose the relevant differences. Accordingly, Fig. 1 focuses on $k \in [0.05, 0.5]$, where architectural differences are most pronounced. Further, as already discussed in A9, the strict low-/middle-/high-order intervals suggested by the reviewer are not defined in Deng et al. (2022) and do not constrain our choice of plotting range, which is why we maintain the current zoomed view.

---

### Meta-Review · Area_Chair_RZNH · 2026-01-05

**Summary:**

This paper proposes to incorporate Shapley value–based interaction regularization into model training to control explanation structure as well as reasoning style across CNN and Transformer architectures. Reviewers found the empirical observations of different interaction patterns between CNN and transformers to interesting, and appreciated the idea of regularizing interpretability during training. However, there are consistent concerns that limit the paper’s contribution. In particular, the proposed regularization closely follows prior work (e.g., Deng et al., 2022) and the novelty largely lies in empirical analysis rather than methodological or theoretical contributions. The claimed theoretical connection between explanation scale and interaction strength is not well supported. Reviewers also noted internal inconsistencies in how interpretability benefits are framed, limited clarity in figures and terms, and insufficient motivation for how controllability relates to robustness or model performance. While the idea is promising, the current version does not clearly demonstrate substantive new contributions over existing methods. Given these concerns, I recommend rejecting the paper in its current form.

**Reviewer Concerns:**

The rebuttal addressed several clarification-related questions, but key concerns regarding limited novelty, weak theoretical justification, and insufficient motivation or evidence for the practical benefits of controllability remain addressed.

**Reviewer Scores:**

Reviewer wwJS lowered their score during the discussion due to concerns about limited contribution over existing work (Deng et al., 2022). Similar concerns regarding limited novelty and insufficient motivation were shared by most reviewers. As a result, I do not expect the final evaluations to be positive.

---

### Decision · Program_Chairs · 2026-01-26

Reject